# Forces are not Enough: Benchmark and Critical Evaluation for Machine Learning Force Fields with Molecular Simulations

## Abstract

Molecular dynamics (MD) simulation techniques are widely used for various natural science applications. Increasingly, machine learning (ML) force field (FF) models begin to replace *ab-initio* simulations by predicting forces directly from atomic structures. Despite significant progress in this area, such techniques are primarily benchmarked by their force/energy prediction errors, even though the practical use case would be to produce realistic MD trajectories. We aim to fill this gap by introducing a novel benchmark suite for ML MD simulation. We curate representative MD systems, including water, organic molecules, peptide, and materials, and design evaluation metrics corresponding to the scientific objectives of respective systems. We benchmark a collection of state-of-the-art (SOTA) ML FF models and illustrate, in particular, how the commonly benchmarked force accuracy is not well aligned with relevant simulation metrics. We demonstrate when and how selected SOTA methods fail, along with offering directions for further improvement. Specifically, we identify stability as a key metric for ML models to improve. Our benchmark suite comes with a comprehensive open source codebase for training and simulation with ML FFs to facilitate further work.

## 1 Introduction

Molecular Dynamics (MD) simulations provide atomistic insights into physical phenomena in materials and biological systems. Such simulations are typically based on force fields (FFs) that characterize the underlying potential energy surface (PES) of the system and then use Newtonian forces to simulate long trajectories (Frenkel & Smit, 2001). The PES itself is challenging to compute and would ideally be done through quantum chemistry which is computationally expensive. Traditionally, the alternative has been parameterized force fields that are surrogate models built from empirically chosen functional forms (Halgren, 1996). Recently, machine learning (ML) force fields (Unke et al., 2021b) have shown promise to accelerate MD simulations by orders of magnitude while being quantum chemically accurate. The evidence supporting the utility of ML FFs is often based on their accuracy in reconstituting forces across test cases (Faber et al., 2017). The evaluations invariably do not involve simulations. **However, we show that force accuracy alone does not suffice for effective simulation (Figure 1).**

MD simulation not only describes microscopic details on how the system evolves, but also entails macroscopic observables that characterize system properties. Calculating meaningful observables often requires long simulations to sample the underlying equilib-

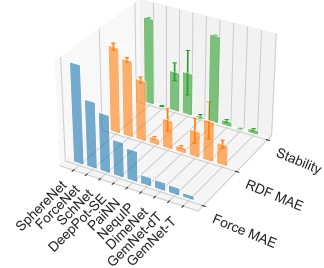

Figure 1: Results on water-10k. Models sorted by force mean absolute error (MAE) in descending order. High stability and low RDF MAE are better. Performance in force error does not align with simulation-based metrics.

rium distribution. These observables are designed to be predictive of material properties such as diffusivity in electrolyte materials (Webb et al., 2015), and reveal detailed physical mechanisms, such as the folding kinetics of protein dynamics (Lane et al., 2011). Although these observables are critical products of MD simulations, systematic evaluations have not been sufficiently studied in existing

literature. To gain insight into the performance of existing models in a simulation setting, we propose a series of simulation-based benchmark protocols. Compared to the popular multistep prediction task in the learned simulator community (Sanchez-Gonzalez et al., 2020), MD observables focus on distributional quantities. The exact recovery of the trajectories given the initial conditions is not the ultimate goal.

Evaluating learned models through MD simulations requires careful design over **the selection of systems, the simulation protocol, and the evaluation metrics**: (1) A benchmark suite should cover diverse and representative systems to reflect the various challenges in different MD applications. (2) Simulations can be computationally expensive. An ideal benchmark needs to balance the cost of evaluation and the complexity of the system so that meaningful metrics can be obtained with reasonable time and computation. (3) Selected systems should be well studied in the simulation domain, and chosen metrics should characterize the system's important degrees of freedom or geometric features.

Are current state-of-the-art (SOTA) ML FFs capable of simulating a variety of MD systems? What might cause a model to fail in simulations? In this paper, our aim is to answer these questions with a novel benchmark study. The contributions of this paper include:

- We introduce a novel benchmark suite for ML MD simulation with simulation protocols and quantitative metrics. We perform extensive experiments to benchmark a collection of SOTA ML models. We provide a complete codebase for training and simulating MD with ML FFs to lower the barrier to entry and facilitate future work.

- We show that many existing models are inadequate when they are evaluated on simulation-based benchmarks, even when they show accurate force prediction (as shown in Figure 1).

- By performing and analyzing MD simulations, we summarize common failure modes and discuss the causes and potential solutions to motivate future research.

## 2 RELATED WORK

**ML force fields** learn the potential energy surface (PES) from the data by applying expressive regressors such as kernel methods (Chmiela et al., 2017) and neural networks on symmetry-preserving representations of atomic environments (Behler & Parrinello, 2007; Khorshidi & Peterson, 2016; Smith et al., 2017; Artrith et al., 2017; Unke & Meuwly, 2018; Zhang et al., 2018b;a; Kovács et al., 2021; Thölke & De Fabritiis, 2021; Takamoto et al., 2022). Recently, graph neural network architectures (Gilmer et al., 2017; Schütt et al., 2017; Gasteiger et al., 2020; Liu et al., 2021) have gained popularity as they provide a systematic strategy for building many-body correlation functions to capture highly complex PES. In particular, equivariant representations have been shown powerful in representing atomic environments (Satorras et al., 2021; Thomas et al., 2018; Qiao et al., 2021; Schütt et al., 2021; Batzner et al., 2022; Gasteiger et al., 2021; Liao & Smidt, 2022), leading to significant improvements in benchmarks such as MD17 and OC22/20. Some works presented simulation-based results (Unke et al., 2021a; Park et al., 2021; Batzner et al., 2022; Musaelian et al., 2022) but do not compare different models with simulation-based metrics.

**Existing benchmarks for ML force fields** (Ramakrishnan et al., 2014; Chmiela et al., 2017) mostly focus on force/energy prediction, with small molecules being the most typical systems. The catalyst-focused OC20 (Chanussot et al., 2021) and OC22 (Tran et al., 2022) benchmarks focus on structural relaxation with force computations, where force prediction is part of the evaluation metrics. The structural relaxation benchmark involves relaxation process of around hundreds of steps, and the goal is to predict the final relaxed structure/energy. These tasks do not characterize system properties under a structural ensemble, which requires simulations that are millions of steps long. Several recent works (Rosenberger et al., 2021) have also studied the utility of certain ML FFs in MD simulations. In particular, Stocker et al. 2022 uses GemNet (Gasteiger et al., 2021) to simulate small molecules in the QM7-x (Hoja et al., 2021) dataset, with a focus on simulation stability. Zhai et al. 2022 applies the DeepMD (Zhang et al., 2018a) architecture to simulate water and demonstrates its shortcoming in generalization across different phases. However, existing works focus on a single system and model, without proposing evaluation protocols and quantitative metrics for model comparison. Systematic benchmarks for simulation-based metrics are lacking in the existing literature, which obscures the challenges in applying ML FF for MD applications.

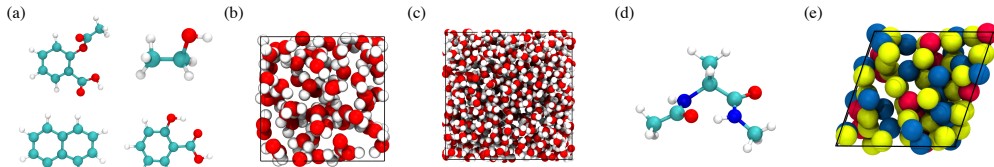

Figure 2: Visualization of the benchmarked systems. (a) MD17 molecules: Aspirin, Ethanol, Naphthalene, and Salicylic acid. (b) 64 water molecules. (c) 512 water molecules. (d) Alanine dipeptide. (e) LiPS.

Table 1: Dataset summary. PBC stands for periodic boundary conditions. *Simulation of alanine dipeptide uses Metadynamics with implicit solvation.

| Dataset | System Type | PBC | #Atoms | Simulation Length | Objective |
|---------|-------------|-----|--------|-------------------|-----------|
| MD17 | Small molecule | ✗ | 9-21 | 300 ps (600k steps) | Interatomic distances |
| Water | liquid | ✓ | 192 | 500 ps (500k steps) | RDF, Diffusivity |
| Water-large | liquid | ✓ | 1536 | 150 ps (150k steps) | RDF, Diffusivity |
| Alanine dipeptide | Peptide | ✗ | 22 | 5 ns (2.5M steps)* | Dihedral angle analysis |
| LiPS | solid-state materials | ✓ | 83 | 50 ps (200k steps) | RDF, Diffusivity |

## 3  PRELIMINARIES

**Training.** An ML FF aims to learn the potential energy surface $\hat{E}(x) \in \mathbb{R}$ as a function of atomic coordinates $\boldsymbol{x} \in \mathbb{R}^{N \times 3}$ ($N$ is the number of atoms), by fitting atom-wise forces $\hat{\boldsymbol{F}}(\boldsymbol{x})$ and energies from a training dataset: $\{\boldsymbol{x}_i, \boldsymbol{F}_i, E_i\}_{i=1}^{N_{\text{data}}}$, where $\boldsymbol{x}_i \in \mathbb{R}^{N \times 3}, \boldsymbol{F} \in \mathbb{R}^{N \times 3}, E \in \mathbb{R}$. For evaluation, the test force prediction accuracy is used as a proxy to quantify the quality of the learned PES. The force field learning protocol has been well established (Unke et al., 2021b).

**MD simulation.** Simulating molecular behaviors requires integrating a Newtonian equation of motion with forces obtained by differentiating $\hat{E}(x)$: $\boldsymbol{F}(\boldsymbol{x}) = -\partial \hat{E}(x)/\partial \boldsymbol{x}$. To mimic desired thermodynamic conditions, an appropriate thermostat and barostat are chosen to augment the equation of motion with extended variables. The simulation produces a time series of positions: $\{x_t \in \mathbb{R}^{N \times 3}\}_{t=0}^{T}$, where $t$ is the temporal order index, and $T$ is the total simulation steps. To evaluate the simulation quality, we propose metrics based-on well-established observables in the respective types of systems. Definitions of benchmarked observables are in Appendix A.

## 4  DATASETS AND METRICS

Popular benchmark datasets, such as MD17, focus on the force prediction task for gas-phase small molecules. However, successes in these tasks are not sufficient evidence for (1) capturing complex interatomic interactions that are critical for condensed phase systems; and (2) recovery of critical simulation observables that cannot be directly indicated by force prediction accuracy. This work focuses on atomic-level MD simulations that manifest complex intermolecular interactions at multiple scales. We choose systems that (1) have been frequently used in force field development (Henderson, 1974); (2) cover diverse MD applications such as materials and biology; and (3) can be simulated within resonable time and compute. Beyond force predictions, we conduct simulations and benchmark observables that reflect the actual simulation quality, along with stability and computational efficiency. The selected systems are summarized in Table 1.

**Quantifying simulation stability.** ML FFs can produce unstable dynamics, as the learned force field may not extrapolate robustly to the undersampled configuration space. As a result, the trajectories can enter nonphysical states that are not meaningful for observable calculations. Therefore, we closely monitor how much the simulated structure deviates from the physical configurations, with the radial distribution function (RDF) for condensed phase systems and bond lengths for flexible molecules. We say that a simulation becomes "unstable" when the deviation exceeds a threshold, which implies sampling of highly nonphysical structures. We then use the time duration for which a model remains stable in simulations to measure its stability. The ensemble statistics are only computed over the stable part of the simulated trajectories: the simulation trajectory before the first occurrence of instability. Details on the stability criterion are included in Appendix A.

Table 2: Models benchmarked in this work. The translation/rotation symmetries are respected by the feature representation at every layer. Number of parameters on the MD17 dataset are reported.

| Model | Symmetry Principle of Geometric Features | Energy Conservation | #Parameters |
|---|---|---|---|
| DeepPot-SE (Zhang et al., 2018b) | E(3)-invariant | ✓ | 1.04M |
| SchNet (Schütt et al., 2017) | E(3)-invariant | ✓ | 0.12M |
| DimeNet (Gasteiger et al., 2020) | E(3)-invariant | ✓ | 2.1M |
| PaiNN (Schütt et al., 2021) | SE(3)-equivariant | ✓ | 0.59M |
| SphereNet (Liu et al., 2021) | E(3)-invariant | ✓ | 1.89M |
| ForceNet (Hu et al., 2021) | Translation-invariant | ✗ | 11.37M |
| GemNet-T (Gasteiger et al., 2021) | E(3)-invariant | ✓ | 1.89M |
| GemNet-dT (Gasteiger et al., 2021) | SE(3)-equivariant | ✗ | 2.31M |
| NequIP (Batzner et al., 2022) | E(3)-equivariant | ✓ | 1.05M |

**MD17** (Chmiela et al., 2017) dataset contains AIMD calculations for eight small organic molecules and is widely used as a force prediction benchmark for ML FFs. We adopt four molecules from MD17 and benchmark the simulation performance. In addition to force error, we evaluate the stability and the distribution of interatomic distances $h(r)$. For each molecule, we randomly sample 9,500 configurations for training and 500 for validation from the MD17 database. We randomly sample 10,000 configurations from the rest of the data for force error evaluation. We perform five simulations of 300 ps for each model/molecule by initializing from 5 randomly sampled testing configurations, with a time step of 0.5 fs, at 500 K temperature, under a Nosé–Hoover thermostat.

**Water** is arguably the most important molecular fluid in biological and chemical processes. Due to its complex thermodynamic and phase behavior, it poses great challenges for molecular simulations. In addition to force error, we evaluate simulation stability and recovery of both equilibrium statistics and dynamical statistics, namely the element-conditioned RDFs and liquid diffusion coefficient. Our dataset consists of 100,000 structures collected every 10 fs from a 1 ns trajectory sampled at equilibrium and a temperature of 300 K. We benchmarked all models with various training+validation dataset sizes (1k/10k/90k randomly sampled structures) and used the remaining 10,000 structures for testing. We performed 5 simulations of 500 ps by initializing from 5 randomly sampled testing configurations, with a time step of 1 fs, at 300 K temperature, under a Nosé–Hoover thermostat. We additionally evaluate model generalization to a larger system of 512 water molecules with 5 simulations of 150-ps.

**Alanine dipeptide** features multiple metastable states, making it a classic benchmark for the development of MD sampling methods and force field development (Head-Gordon et al., 1989; Kaminski et al., 2001). Its geometric flexibility is well represented by the central dihedral (torsional) angles $\phi$ and $\psi$. Our reference data are obtained from simulations with explicit water molecules, with detailed protocols described in Appendix B. For faster simulation, we learn an implicitly solvated FF following a protocol similar to Chen et al. (2021). Our task is more challenging in that it aims to learn the implicitly solvated atomistic FF rather than the implicit solvation correction in Chen et al. (2021). To facilitate accelerated sampling, we apply metadynamics with $\phi$ and $\psi$ as the collective variables. We evaluate force prediction, simulation stability, and free energy surface (FES) reconstruction $F(\phi, \psi)$. Our dataset consists of 50,000 structures dumped every 2 ps from a 100 ns trajectory at a temperature of 300 K. We used 38,000 randomly sampled structures for training, 2,000 for validation, and the rest as a test set. We performed 6 simulations of 5 ns by initializing from six local minima on the FES (Figure 5) with a time step of 2 fs at 300 K, and under a Langevin thermostat to mimic random noise from solvation effects. More information on our simulation protocols can be found in Appendix B.

**LiPS** is a crystalline superionic lithium conductor relevant to battery development and a representative system for MD simulation usage in studying kinetic properties in materials. We adopt this dataset from Batzner et al. 2022, and benchmark all models on their force error, stability, RDF recovery, and Li-ion diffusivity coefficient. The dataset has 25,000 structures in total, from which we use 19,000 randomly sampled structures for training, 1,000 structures for validation, and the rest for computing force error. We conduct 5 simulations of 50 ps by initializing from 5 randomly sampled testing configurations, with a time step of 0.25 fs, at 520 K temperature, under a Nosé–Hoover thermostat.

## 5 EXPERIMENTS

**Benchmarked models.** We adopt the Open Catalyst Project implementation of SchNet (Schütt et al., 2017), DimeNet (Gasteiger et al., 2020), ForceNet (Hu et al., 2021), PaiNN (Schütt et al.,

Table 3: Results on MD17. Darker green color indicates better performance. For all results, force MAE is reported in the unit of [meV/Å], and stability is reported in the unit of [ps]. The distribution of interatomic distances $h(r)$ MAE is unitless. FPS stands for frames per second. For all metrics ($\downarrow$) indicates the lower the better, and ($\uparrow$) indicates the higher the better. Standard deviation from 5 simulations is in subscript for applicable metrics.

| Molecule | Model | DeepPot-SE | SchNet | DimeNet | PaiNN | SphereNet | ForceNet | GemNet-T | GemNet-dT | NequIP |
|---|---|---|---|---|---|---|---|---|---|---|
| Aspirin | Force ($\downarrow$) | 21.0 | 35.6 | 10.0 | 9.2 | 3.4 | 22.1 | 3.3 | 5.1 | 2.3 |
| | Stability ($\uparrow$) | $9_{(15)}$ | $26_{(23)}$ | $54_{(12)}$ | $159_{(121)}$ | $141_{(54)}$ | $182_{(144)}$ | $72_{(50)}$ | $192_{(132)}$ | $300_{(0)}$ |
| | $h(r)$ ($\downarrow$) | $0.65_{(0.47)}$ | $0.36_{(0.57)}$ | $0.04_{(0.00)}$ | $0.04_{(0.01)}$ | $0.03_{(0.00)}$ | $0.56_{(0.15)}$ | $0.04_{(0.02)}$ | $0.04_{(0.01)}$ | $0.02_{(0.00)}$ |
| | FPS ($\uparrow$) | 88.0 | 108.9 | 20.6 | 85.8 | 17.5 | 137.3 | 28.2 | 56.8 | 8.4 |
| Ethanol | Force | 8.9 | 16.8 | 4.2 | 5.0 | 1.7 | 14.9 | 2.1 | 1.7 | 1.3 |
| | Stability | $300_{(0)}$ | $247_{(106)}$ | $26_{(10)}$ | $86_{(109)}$ | $33_{(16)}$ | $300_{(0)}$ | $169_{(98)}$ | $300_{(0)}$ | $300_{(0)}$ |
| | $h(r)$ | $0.09_{(0.00)}$ | $0.21_{(0.11)}$ | $0.15_{(0.03)}$ | $0.15_{(0.08)}$ | $0.13_{(0.03)}$ | $0.86_{(0.05)}$ | $0.10_{(0.02)}$ | $0.09_{(0.00)}$ | $0.08_{(0.00)}$ |
| | FPS | 101.0 | 112.6 | 21.4 | 87.3 | 30.5 | 141.1 | 27.1 | 54.3 | 8.9 |
| Naphthalene | Force | 13.4 | 22.5 | 5.7 | 3.8 | 1.5 | 9.9 | 1.5 | 1.9 | 1.1 |
| | Stability | $246_{(109)}$ | $18_{(2)}$ | $85_{(68)}$ | $300_{(0)}$ | $6_{(3)}$ | $300_{(0)}$ | $8_{(2)}$ | $25_{(10)}$ | $300_{(0)}$ |
| | $h(r)$ | $0.11_{(0.00)}$ | $0.09_{(0.00)}$ | $0.10_{(0.01)}$ | $0.13_{(0.00)}$ | $0.14_{(0.04)}$ | $1.02_{(0.00)}$ | $0.13_{(0.00)}$ | $0.12_{(0.01)}$ | $0.12_{(0.01)}$ |
| | FPS | 109.3 | 110.9 | 19.1 | 92.8 | 18.3 | 140.2 | 27.7 | 53.5 | 8.2 |
| Salicylic Acid | Force | 14.9 | 26.3 | 9.6 | 6.5 | 2.6 | 12.8 | 4.0 | 4.0 | 1.6 |
| | Stability | $300_{(0)}$ | $300_{(0)}$ | $73_{(82)}$ | $281_{(37)}$ | $36_{(16)}$ | $1_{(0)}$ | $26_{(24)}$ | $94_{(109)}$ | $300_{(0)}$ |
| | $h(r)$ | $0.03_{(0.00)}$ | $0.03_{(0.00)}$ | $0.06_{(0.02)}$ | $0.03_{(0.00)}$ | $0.06_{(0.02)}$ | $0.35_{(0.00)}$ | $0.08_{(0.04)}$ | $0.07_{(0.03)}$ | $0.03_{(0.00)}$ |
| | FPS | 94.6 | 111.7 | 19.4 | 90.5 | 21.4 | 143.2 | 28.5 | 52.4 | 8.4 |

Figure 3: Head-to-head comparison of force MAE vs. Stability and $h(r)$ MAE on MD17 molecules. Models are on the x-axis and are sorted according to force error in descending order. High stability and low $h(r)$ MAE mean better performance. Error bars indicate 95% confidence intervals.

2021), GemNet-T/dT (Gasteiger et al., 2021), and the official implementation of DeepPot-SE (Zhang et al., 2018b), SphereNet (Liu et al., 2021), and NequIP (Batzner et al., 2022). A summary of all benchmarked models are in Table 2. These models have been popular in previous benchmark studies for force/energy prediction. They use different representations for atomistic interactions and respect different levels of euclidean symmetry. We follow all original hyperparameters introduced in the respective papers and only make minimal adjustments when the training is unstable. More details on the hyperparameters can be found in Appendix C.

**Key observations.** We make two key observations as evidenced in the experimental results:

1. Despite being widely used, force prediction is not sufficient for evaluating ML FFs. It generally does not align with simulation stability and performance in estimating ensemble properties.

2. While often neglected, stability can be a major bottleneck for ML FFs. Lower force error and more training data does not necessarily give rise to more stable simulations, suggesting stability as a fundamental consideration for comparison and model design.

We next go through experimental results of all four datasets in detail to demonstrate the key observations while making other observations.

**MD17.** As shown in Table 3, more recent models that lie on the right side of the table generally achieve a lower force error, but may lack stability. Figure 3 selects results from Table 3 to demonstrate the non-aligned trends of force prediction performance vs. simulation performance, which supports **key observation 1**. SphereNet and GemNet-T/dT can attain a very low force error for all four molecules, but often collapse before the simulation finishes. This observation constitutes **key observation 2**. We note that although the stable portion of simulated trajectories produced by SphereNet and GemNet-T/dT can recover the $h(r)$ relatively accurately, stability will become a bigger issue when the statistics

Table 4: Results on Water-10k. RDF MAE is unit-less. Diffusivity is computed by averaging 5 runs from 5 random initial configurations and its MAE is reported in the unit of $[10^{-9}\ \mathrm{m}^2/\mathrm{s}]$. The reference diffusivity coefficient is $2.3 \times 10^{-9}\ \mathrm{m}^2/\mathrm{s}$.

| | DeepPot-SE | SchNet | DimeNet | PaiNN | SphereNet | ForceNet | GemNet-T | GemNet-dT | NequIP |
|---|---|---|---|---|---|---|---|---|---|
| Force ($\downarrow$) | 5.8 | 9.5 | 1.4 | 5.1 | 16.1 | 10.9 | 0.7 | 1.3 | 1.5 |
| Stability ($\uparrow$) | $247_{(147)}$ | $232_{(59)}$ | $30_{(10)}$ | $12_{(13)}$ | $500_{(0)}$ | $7_{(3)}$ | $25_{(7)}$ | $7_{(3)}$ | $500_{(0)}$ |
| RDF$_{(O,O)}$ ($\downarrow$) | $0.07_{(0.01)}$ | $0.63_{(0.04)}$ | $0.27_{(0.15)}$ | $0.30_{(0.14)}$ | $0.89_{(0.04)}$ | $0.79_{(0.03)}$ | $0.22_{(0.05)}$ | $0.42_{(0.22)}$ | $0.06_{(0.02)}$ |
| RDF$_{(H,H)}$ ($\downarrow$) | $0.06_{(0.02)}$ | $0.30_{(0.02)}$ | $0.18_{(0.08)}$ | $0.21_{(0.09)}$ | $0.40_{(0.01)}$ | $0.55_{(0.01)}$ | $0.16_{(0.03)}$ | $0.35_{(0.25)}$ | $0.05_{(0.01)}$ |
| RDF$_{(H,O)}$ ($\downarrow$) | $0.19_{(0.05)}$ | $0.57_{(0.04)}$ | $0.21_{(0.04)}$ | $0.29_{(0.12)}$ | $1.14_{(0.03)}$ | $1.34_{(0.03)}$ | $0.20_{(0.04)}$ | $0.42_{(0.27)}$ | $0.27_{(0.07)}$ |
| Diffusivity ($\downarrow$) | 0.04 | 1.90 | - | - | 2.23 | - | - | - | 0.18 |
| FPS ($\downarrow$) | 91.0 | 78.9 | 17.9 | 71.8 | 3.1 | 67.6 | 11.3 | 33.7 | 3.9 |

Figure 4: Comparison of force MAE vs. stability (Left), force MAE vs. RDF MAE (Middle), and force MAE vs. Diffusivity MAE (Right) on the water benchmark. Each model is trained with three dataset sizes. The color of a point indicates the model identity, while the point size indicates the training dataset size (small: 1k, medium: 10k, large: 90k). Metrics infeasible to extract from certain model/dataset size (e.g., Diffusivity for unstable models) are not included.

of interest require long simulations, as demonstrated in other experiments. On the other hand, despite having a relatively high force error, DeepPot-SE performs very well on simulation-based metrics on all molecules except for Aspirin (Figure 3). With the highest molecular weight, Aspirin is indeed the hardest task in MD17 in the sense that all models attain high force prediction errors on it. PaiNN also attains competitive simulation performance while its force error is not among the best.

We further observe that good stability does not imply accurate recovery of trajectory statistics. Although ForceNet remains stable for Ethanol and Naphthalene, the extracted $h(r)$ deviates a lot from the reference (Table 3), indicating that ForceNet does not learn the underlying PES correctly, possibly due to its lack of energy conservation and rotational equivariance. Overall, NequIP is the best-performing model on MD17. It achieves the best performance in both force prediction and simulation-based metrics for all molecules while requiring the highest computational cost. More detailed results on MD17 including a study on stability's relation with training epochs and individual $h(r)$ are included in Appendix C.

**Water.** Under different challenges posed by a condensed phase system, **key observation 1 and 2** are still evident according to Table 4: GemNet-T/dT and DimeNet are the top-3 models in terms of force prediction, but all lack stability. The water diffusivity coefficient requires long (100 ps in our experiments) trajectories to estimate and thus cannot be extracted for unstable models. Like MD17, DeepPot-SE does not achieve the best force prediction performance but demonstrates decent stability and highly accurate recovery of simulation statistics. Interestingly, SphereNet has high force error but is highly stable. However, the properties are not accurately recovered.

Figure 4 further compares model performance with different training dataset sizes. **Key observation 1 and 2** are clearly shown: Models located on the left of each scatter plot have very low force error but may have poor stability or high error in simulation statistics. More specifically, although more training data almost always improve force prediction performance, its effect on simulation performance is not entirely clear. On the one hand, GemNet-T/dT, DimeNet, and ForceNet are not stable even when under the highest training data budget. On the other hand, we observe a clear

Table 5: Results on alanine dipeptide. #Finished is the number of simulations stable for 5 ns. MAE of $F(\phi)$ and $F(\psi)$ are reported in the unit of [kJ/mol].

| | DeepPot-SE | SchNet | DimeNet | ForceNet | GemNet-T | GemNet-dT | NequIP |
|---|---|---|---|---|---|---|---|
| Force ($\downarrow$) | 272.1 | 217.0 | 239.0 | 284.7 | 233.5 | 219.7 | 215.6 |
| #Finished ($\uparrow$) | 0/6 | 0/6 | 0/6 | 0/6 | 0/6 | 0/6 | 5/6 |
| Stability ($\uparrow$) | $0_{(0)}$ | $0_{(0)}$ | $0_{(0)}$ | $0_{(0)}$ | $18_{(27)}$ | $0_{(0)}$ | $4168_{(1860)}$ |
| $F(\phi)$ ($\downarrow$) | - | - | - | - | - | - | $108_{(2)}$ |
| $F(\psi)$ ($\downarrow$) | - | - | - | - | - | - | $126_{(4)}$ |
| FPS ($\uparrow$) | 54.3 | 42.4 | 12.1 | 99.1 | 15.0 | 36.5 | 8.3 |

Table 6: Results on LiPS. Li-ion Diffusivity coefficient is computed by averaging 5 runs from 5 random initial configurations. The reference Li-ion diffusivity coefficient is $1.35 \times 10^{-9}$ m$^2$/s.

| | DeepPot-SE | SchNet | DimeNet | ForceNet | GemNet-T | GemNet-dT | NequIP |
|---|---|---|---|---|---|---|---|
| Force ($\downarrow$) | 40.5 | 28.8 | 3.2 | 12.8 | 1.3 | 1.4 | 3.7 |
| Stability ($\uparrow$) | $4_{(3)}$ | $50_{(0)}$ | $48_{(4)}$ | $26_{(8)}$ | $50_{(0)}$ | $50_{(0)}$ | $50_{(0)}$ |
| RDF ($\downarrow$) | $0.27_{(0.15)}$ | $0.04_{(0.00)}$ | $0.05_{(0.01)}$ | $0.51_{(0.08)}$ | $0.04_{(0.00)}$ | $0.04_{(0.00)}$ | $0.04_{(0.01)}$ |
| Diffusivity ($\downarrow$) | - | 0.38 | 0.30 | - | 0.24 | 0.28 | 0.34 |
| FPS ($\uparrow$) | 66.1 | 35.2 | 14.8 | 72.1 | 16.9 | 43.5 | 8.2 |

improvement of DeepPot-SE when more training data is used. NequIP is again the best performing model, achieving very low force error, excellent stability, and accurate recovery of ensemble statistics, even under the lowest data budget of 1,000 training+validation structures. However, when the training dataset is sufficiently large (90k), DeepPot-SE has equally good results as NequIP while being more than 20 times faster – dataset size also influences the model of choice for a certain dataset. More results, including tables for water-1k/90k, a study on model size, and generalization to Water-large are included in Appendix C.

**Alanine dipeptide** poses unique challenges in sampling different metastable states separated by high free energy barriers. Table 5 shows all models have high force errors due to the random forces introduced by the lack of explicit account of water molecules. Although the force errors are in the same order of magnitude, all models except NequIP fail to simulate stably. The FES reconstruction task requires stable simulation for the entire 5 ns. NequIP is the only model that manages to finish five simulations out of six but produces inaccurate statistics. All other models are not stable enough to produce meaningful results. We further analyze the results on this task in Section 6.

**LiPS.** Compared to flexible molecules and liquid water, this solid material system features slower kinetics. From Table 6 we observe that most models are capable of finishing 50-ns simulations stably. In this dataset, the performance on diffusivity estimation and force prediction align well. We observe that both GemNet-T and GemNet-dT show excellent force prediction, stability, and recovery of observables, while GemNet-dT is 2.6 times faster. The better efficiency comes from the direct prediction of atomic forces $\boldsymbol{F}$ instead of taking the derivative $\boldsymbol{F} = \partial E/\partial \boldsymbol{x}$, which also makes GemNet-dT not energy-conserving – a potential issue we further discuss in Section 6.

**Implications on model architecture.** More recent models utilizing SE(3)/E(3)-equivariant representations and operations such as GemNet-dT and NequIP are more expressive and can capture interatomic interactions more accurately. This is reflected by their very low force error and accurate recovery of ensemble statistics when not bottlenecked by stability. Moreover, NequIP shows that excellent accuracy and stability can be simultaneously achieved. The stability may come from parity-equivariance and the explicit architecture in manipulating higher-order geometric tensors. We believe further investigations into the extrapolation behavior induced by different equivariant geometric representations and operations (Batatia et al., 2022) is a fruitful direction in designing more powerful ML FFs.

## 6    FAILURE MODES: CAUSES AND FUTURE DIRECTIONS

**A case study on alanine dipeptide simulation.** NequIP achieves decent performance on all our tasks but fails on alanine dipeptide. It is also the only model that can simulate stably for 5 ns. Figure 5 (a) demonstrates how NequIP fails to reconstruct the FES: it does not manage to sample much of the

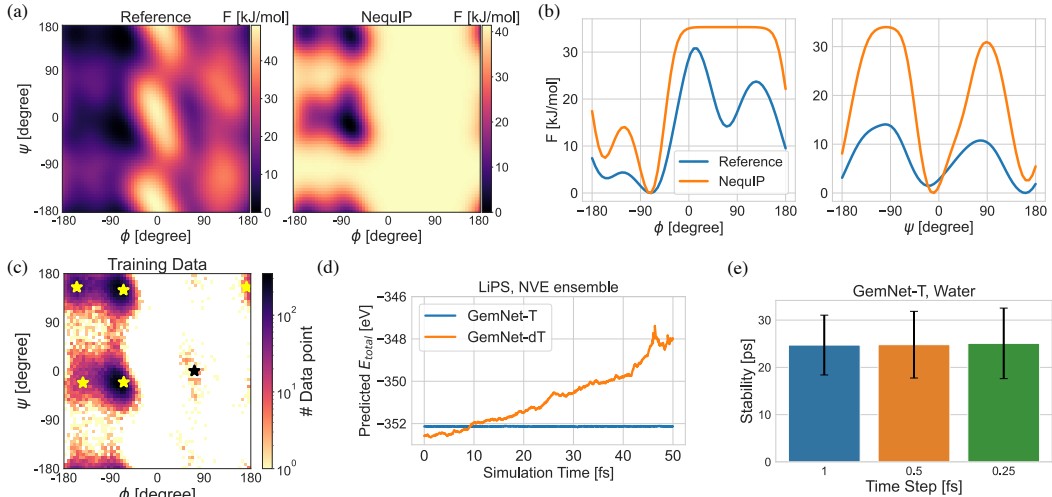

Figure 5: (a) Ramachandran plots of the alanine dipeptide FES reconstructed from 5-ns reference vs. 5-ns NequIP simulation, both using MetaDynamics. (b) $F(\phi)$ and $F(\psi)$ of alanine dipeptide extracted from reference simulation vs. from NequIP simulation. (c) $(\phi, \psi)$ distribution of the alanine dipeptide training dataset. The six initialization points are marked with stars. NequIP fails to remain stable when the simulation starts from the point marked with black color. (d) Model-predicted total energy as a function of simulation time when simulating the LiPS system using the NVE ensemble. (e) On water-10k, stability does not improve when the time step is reduced for GemNet-T.

transition regions and the configuration space with $\phi \in [0, 180°]$. Figure 5 (b) demonstrates the reconstructed FES, which significantly deviates from the reference. This failure can be partially explained by Figure 5 (c), the training data distribution produced by the reference potential. The relatively high-energy (low-density) regions are exactly those that are not reachable by NequIP. Even though our MD trajectory is well-equilibrated, the relative difference in populations of different meta-stable states creates data imbalance, making it more challenging for the model to learn PES for higher-energy configurations where density is relatively low. In our experiments, we observe that simulations starting from the low-density meta-stable state (e.g., black star marked in Figure 5 (c)) tend to fail. This implies that generalization across different regions in the conformational space is an important challenge for ML FFs. To prevent ML FFs from sampling nonphysical regions, which is a common precursor to failed simulation (Figure 6), one can deliberately include distorted and off-equilibrium geometries in the training data to improve model robustness(Stocker et al., 2022). Alternatively, one can resort to active learning (Wang et al., 2020b; Vandermause et al., 2020; Schwalbe-Koda et al., 2021) to acquire new data points based on model uncertainty.

**Energy conservation.** Models that directly predict forces may not conserve energy. Figure 5 (d) demonstrates the evolution of model-predicted total energy for selected models on LiPS, in a micro-canonical (NVE) ensemble. The energy of an isolated system in the NVE ensemble is in principle conserved. We observe that GemNet-T conserves energy, whereas GemNet-dT fails to conserve the predicted total energy. The existence of non-conservative forces breaks the time reversal symmetry and, therefore, may not be able to reach equilibrium for observable calculation. However, in our experiment, GemNet-dT performs well on the LiPS dataset when coupled with a thermostat. Previous works (Kolluru et al., 2022) also found that energy conservation is not required for SOTA performance on OC20. The usability of non-conservative FFs in simulations requires further careful investigations.

**Simulation instability** is a major bottleneck for highly accurate models such as GemNet-T to fail on several simulation tasks. Moreover, in our water experiments, we find a larger amount of training data does not resolve this issue for GemNet-T/dT and DimeNet (Figure 4). We further experiment with smaller simulation time steps for GemNet-T on water (Figure 5 (e)), but stability still does not improve. On the other hand, Stocker et al. (2022) demonstrates that the stability of GemNet improves with larger training sets on QM7-x, which includes high-energy off-equilibrium geometries obtained from normal mode sampling. We hypothesize that including these distorted geometries may improve the model's robustness against going into nonphysical configurations. We also observe that

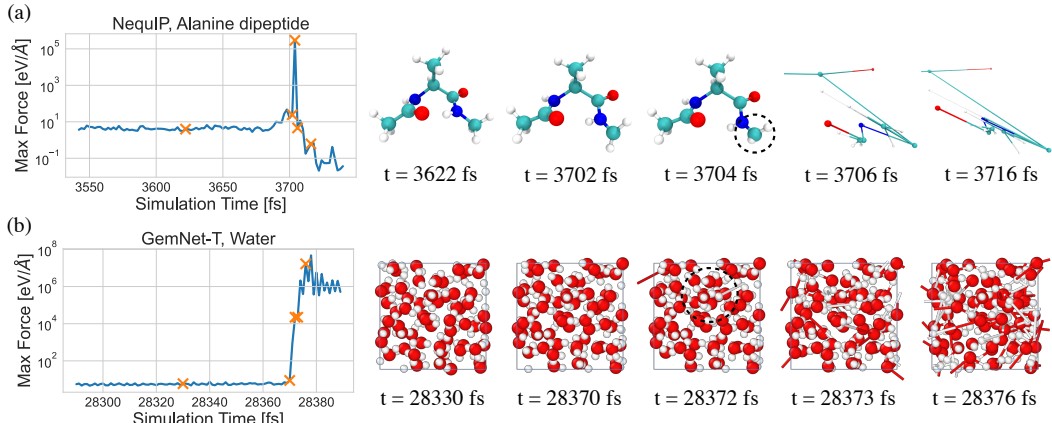

Figure 6: Examples of simulation collapse when applying (a) NequIP to alanine dipeptide and (b) GemNet-T to water. The y-axis shows the maximum force observed on any atom in the system at a certain time step. An orange cross indicates visualized time steps. Notable nonphysical regions are circled. The collapse usually happens within a very short period of time after the initial local errors.

the simulation can collapse within a short time window after a long period of stable simulation, as visualized in Figure 6. In both cases, we observe that the nonphysical configurations first emerge at local regions (circled), which cascade to the entire system very quickly as extremely large forces are being produced and subsequently integrated into the dynamics. At the end of the visualization, the bonds in the alanine dipeptide system are broken. Therefore, the local-descriptor-based NequIP model predicts very small forces. For the water system, the particles are packed in a finite periodic box. The nonphysical configurations exhibit incorrect coordination structures and extremely large forces. Regarding stability, past works found adding noise paired with a denoising objective during training helpful in improving out-of-distribution test performance on OC20 (Godwin et al., 2021), and in stabilizing learned simulations (Sanchez-Gonzalez et al., 2020). Another relevant line of work in coarse-grained MD simulation has studied regularization with an empirical "prior energy" (Wang et al., 2019b) and post-prediction refinement (Fu et al., 2022) to battle simulation instability.

## 7  CONCLUSION AND OUTLOOK

We have introduced a diverse suite of MD simulation tasks and conducted a thorough comparison of SOTA ML FFs to reveal novel insights into ML for MD simulation. As shown in our experiments, benchmarking only force error is not sufficient, and simulation-based metrics should be used to reflect the practical utility of a model. We demonstrate case studies on the failure of existing training schemes/models to better understand their limitations. The performance of a model can be highly case-dependent. For more challenging MD systems, more expressive atomistic representations may be required. For example, recent work has explored non-local descriptors (Kabylda et al., 2022) aiming at capturing long-range interactions in large molecules. Strictly local equivariant representations (Musaelian et al., 2022) are studied for very large systems where computational scalability is critical. New datasets (Eastman et al., 2022) and benchmarks have been playing an important role in inspiring future work. This work focuses on atomic-level simulations. Simulating large systems such as polymers and proteins at atomic resolution would be too expensive for existing models. Learning coarse-grained MD (Wang et al., 2019b; Wang & Gómez-Bombarelli, 2019; Fu et al., 2022) is another avenue to accelerate MD at larger length/time scales.

The possibility of ML in advancing MD simulation is not limited to ML force fields. Enhanced sampling methods enable fast sampling of rare events and have been augmented with ML techniques (Schneider et al., 2017; Sultan et al., 2018; Holdijk et al., 2022). Differentiable simulations (Schoenholz & Cubuk, 2020; Wang et al., 2020a; Doerr et al., 2021; Ingraham et al., 2018; Greener & Jones, 2021) offer a principled way of learning the force field by directly training the simulation process to reproduce experimental observables (Wang et al., 2020a; 2022; Thaler & Zavadlav, 2021). We hope our datasets and benchmarks will encourage future developments in all related aspects to push the frontier in ML for MD simulations.

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

## A    EVALUATION METRICS

**Observables.** From the time series observations of positions and velocities, observables $O(x_t)$ can be computed to characterize the state of the system at different granularities. Under the ergodic hypothesis, the time averages of the simulation observables converge to distributional averages under the Gibbs measure: $\langle O \rangle = \frac{1}{T} \lim_{T \to \infty} \sum_t^T O(x_t) = \int dx \, p(x) \, O(x)$, where $p(x) \propto \exp(-\frac{\hat{E}(x)}{k_B T})$ with $T$ as the bath temperature and $k_B$ as the Boltzmann constant. Calculations of such observables require the system to reach equilibrium. Simulation observables connect simulations to experimental measurements and are predictive of macroscopic properties of matter. Common observables include radial distribution functions (RDFs), virial stress tensor, mean-squared displacement (MSD), etc.

**Distribution of interatomic distances** is a low-dimensional description of the 3D structure and has been studied in previous work (Zhang et al., 2018a). For a given configuration $\boldsymbol{x}$, the distribution of interatomic distances $h(r)$ is computed with:

$$h(r) = \frac{1}{N(N-1)} \sum_i^N \sum_{j \neq = i}^N \delta(r - ||\boldsymbol{x}_i - \boldsymbol{x}_j||) \tag{1}$$

where $r$ is the distance from a reference particle; $N$ is the total number of particles; $i, j$ indicates the pairs of atoms that contribute to the distance statistics; $\delta$ is the Dirac Delta function to extract value distributions. To calculate the ensemble average, $h(r)$ is calculated and averaged over frames from equilibrated trajectories. For learned simulations, we compute $h(r)$ using only the stable part of the simulation.

**RDF.** As one of the most informative simulation observables, the radial distribution function (RDF) describes the structural/thermodynamic properties of the system and is also experimentally measurable. It has been widely used in force field development (Henderson, 1974). By definition, the RDF describes how density varies as a function of distance from a particle (illustrated in Figure 7 (a)). For a given configuration $\boldsymbol{x}$, the RDF can be computed with the following formula:

$$\text{RDF}(r) = \frac{1}{4\pi r^2} \frac{1}{N\rho} \sum_i^N \sum_{j \neq = i}^N \delta(r - ||\boldsymbol{x}_i - \boldsymbol{x}_j||) \tag{2}$$

where $r$ is the distance from a reference particle; $N$ is the total number of particles; $i, j$ indicates the pairs of atoms that contribute to the distance statistics; $\rho$ is the density of the system; $\delta$ is the Dirac Delta function to extract value distributions. To calculate the ensemble average, $\text{RDF}(r)$ is calculated and averaged over frames from equilibrated trajectories. The final RDF MAE is then calculated by integrating r:

$$\text{MAE}_{\text{RDF}} = \int_{r=0}^{\infty} |\langle \text{RDF}(r) \rangle - \langle \hat{\text{RDF}}(r) \rangle| dr \tag{3}$$

where $\langle \cdot \rangle$ is the averaging operator, $\langle \text{RDF}(r) \rangle$ is the reference equilibrium RDF, and $\langle \hat{\text{RDF}}(r) \rangle$ is the model-predicted RDF.

**Diffusivity coefficient** is a major property of study in many previous work (Batzner et al., 2022). The diffusivity coeffcient $D$ quantifies the time-correlation of the translational displacement (illustrated in Figure 7 (b)), and can be computed from the mean square displacement:

$$D = \lim_{t \to \infty} \frac{1}{6t} \frac{1}{N'} \sum_{i=1}^{N'} |\boldsymbol{x}_i(t) - \boldsymbol{x}_i(0)|^2 \tag{4}$$

where $x_i(t)$ is the coordinate of particle $i$ at time $t$, and $N'$ is the number of particles being tracked in the system. For the water system we monitor the liquid diffusivity coefficient and track all 64 oxygen atoms. For the LiPS system, we monitor Li-ion Diffusivity and track all 27 Li-ions. As the definition implies, $D$ is a quantity that converges with longer simulation time. Accurate recovery of $D$ requires sufficient long trajectories sampled from the Hamiltonian with ML FFs. In this paper,

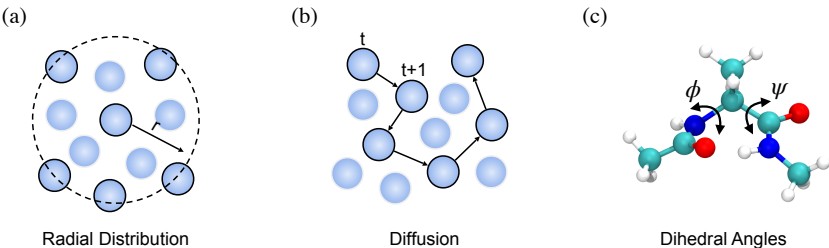

(a)                          (b)                          (c)

Radial Distribution          Diffusion          Dihedral Angles

Figure 7: Illustrations regarding benchmarked metrics.

we only compute diffusivity for stable trajectories of at least 100 ps for water and 40 ps for LiPS. As we simulate multiple random runs for each system/model, we average the diffusivity coefficient extracted from each valid trajectory to obtain the final prediction.

**Free energy surface.** Given the probability distributions over configurations $p(\boldsymbol{x})$ and a chosen geometric coordinate $\boldsymbol{\xi}$ transformed from $\boldsymbol{x}$. Based on the marginalized density $p(\boldsymbol{\xi})$, the free energy (Tuckerman, 2010) can be calculated from the following:

$$F(\boldsymbol{\xi}) = -k_B T \ln p(\boldsymbol{\xi}) \tag{5}$$

In the specific case of alanine dipeptide, there are two main conformational DOF: dihedral angle $\phi$ of $C - N - C_\alpha - C$ and dihedral angle $\psi$ of $N - C_\alpha - C - N$ (illustrated in Figure 7 (c)). Therefore, the FES w.r.t $\phi$ and $\psi$ is the most physically informative. We propose our quantitative metric $\mathrm{MAE}_{F(\phi), F(\psi)}$ based on the absolute error in reconstructing the FES along the $\phi$ and $\psi$ coordinates. We integrate the absolute difference between the reference free energy $F$ and the model predicted $\hat{F}$ from $[-\pi, \pi)$:

$$\mathrm{MAE}_{F(\phi)} = \int_{\phi=-\pi}^{\pi} |F(\phi) - \hat{F}(\phi)| d\phi \tag{6}$$

$$\mathrm{MAE}_{F(\psi)} = \int_{\psi=-\pi}^{\pi} |F(\psi) - \hat{F}(\psi)| d\psi \tag{7}$$

**Stability criterion.** Abstractly speaking, stability is a notion of staying within physical (low-energy) configuration spaces. Since all MD systems studied in this paper are in equilibrium, practically we keep track of stability by closely monitoring equilibrium statistics. For systems with periodic boundary conditions, we monitor the RDF and say a simulation becomes "unstable" at time $T$ when

$$\int_{r=0}^{\infty} ||\langle \mathrm{RDF}(r) \rangle - \langle \hat{\mathrm{RDF}}_t(r) \rangle_{t=T}^{T+\tau}|| dr > \Delta \tag{8}$$

where $\langle \cdot \rangle$ is the averaging operator, $\tau$ is a short time window, and $\Delta$ is the stability threshold. In this paper we use $\tau = 1$ ps, $\Delta = 3.0$ for water, and $\tau = 1$ ps, $\Delta = 1.0$ for LiPS. For water, we assert unstable if any of the three element-conditioned RDFs: $\mathrm{RDF}_{(O,O)}, \mathrm{RDF}_{(H,H)}, \mathrm{RDF}_{(H,O)}$ exceeds the threshold. For flexible molecules, we keep track of stability through the bond lengths and say a simulation becomes "unstable" at time $T$ when:

$$\max_{(i,j) \in \mathcal{B}} |(||\boldsymbol{x}_i(T) - \boldsymbol{x}_j(T)|| - b_{i,j})| > \Delta \tag{9}$$

where $\mathcal{B}$ is the set of all bonds, $i, j$ are the two endpoint atoms of the bond, and $b_{i,j}$ is the equilibrium bond length. We use $\Delta = 0.5$ Å for both MD17 molecules and alanine dipeptide.

We measure the stability of a learned model through the simulation time it remains stable. For each dataset, the threshold $\Delta$ we adopt is rather relaxed that an "unstable" equilibrium statistic usually indicates the system is already in a highly nonphysical configuration.

**FPS.** All frames per second (FPS) metrics are measured with an NVIDIA Tesla V100-PCIe GPU. We present FPS as a reference for models' computational efficiency but also note that code speed can be affected by many factors, and likely has room for improvement.

**Experimental procedures.** Baseline models are trained on the datasets described in Section 4 according to the experimental settings described in Appendix C. At evaluation time, we simulate MD trajectories using the learned models, with thermostats and simulation length described in Section 4. The simulated trajectories are recorded as time series of atom positions, along with other information including atom types, temperature, total energy, potential energy, and kinetic energy. All observables described in this section can be computed from recorded trajectories. We use the stability criterion described above to find the time step when the systems become "unstable", and only use the trajectory before that time step for the computation of observables. Among the observables, the distribution of interatomic distances and RDF are computed for each frame, and averaged over the entire trajectory. Diffusivity coefficients are computed by averaging over the diffusivity coefficient computed from all applicable time windows where the time window length is predefined to be long enough. For example, for a trajectory of $T$ steps and a time window of $K$ steps, we average over the diffusivity computed from the time windows: $[1, K], [2, K+1], \ldots, [T-K+1, T]$. We use 100 ps as the time window size for water and 35 ps as the time window size for LiPS. We also remove the first 5 ps of the simulated trajectories of LiPS for equilibrium. As we do multiple simulations per model and dataset, we compute the metrics for each trajectory and report the mean and standard deviation. Further details on observable computation can be found in our code submission: `observable.ipynb`.

## B  DATASET DETAILS

The MD17 dataset[1] (Chmiela et al., 2017) and the LiPS dataset[2] (Batzner et al., 2022) are adapted from previous works and are publicly available. The MD17 dataset is generated from path-integral molecular dynamics simulations that incorporate quantum mechanics into the classic molecular dynamics simulations using Feynman path integrals. The LiPS datasets are generated by ab-initio molecular dynamics simulations with a generalized gradient PBE functional and projector augmented wave pseudopotentials. We refer interested readers to the respective papers for more details on the data generation process. The water dataset and alanine dipeptide dataset are generated by ourselves. process.

**Water.** Our water dataset is generated from molecular dynamics simulations of a simple classical water model, namely, the flexible version of the Extended Simple Point Charge water model (SPC/E-fw) (Wu et al., 2006) at temperature $T = 300$ K and pressure $P = 1$ atm. For this model, the interaction parameters (e.g., O-H bond stretch and H-O-H bond angles), are parameterized to match extensive experimental properties such as the self-diffusion and dielectric constants at bulk phase. This classical model has been well-studied in previous work (Wu et al., 2006; Yue et al., 2021) and has shown reasonable predictions of the physical properties of liquid water. It provides a computationally inexpensive way to generate a large amount of training data. The experience and knowledge gained from the benchmark based on the simple model can be readily extended to systems with higher accuracy, such as the *ab-initio* models.

**Alanine dipeptide.** Our dataset is generated from the MD simulation of an alanine dipeptide molecule solvated in explicit water (1164 water molecules) performed in GROMACS (Abraham et al., 2015) using the AMBER-03 (Ponder & Case, 2003b) force-field. In the AMBER-03 force field, the potential energy parameters such as van der Waals and electrostatics are mostly derived from quantum mechanical methods with minor optimization on the bonded parameters to reproduce the experimental vibrational frequencies and structures (Cornell et al., 1996; Ponder & Case, 2003a). The NPT ensemble is applied in simulations, with hydrogen bond length constraints using LINear Constraint Solver (LINCS) and a time step of 2 fs. The temperature and pressure of the system are controlled at $T = 300$ K and $P = 1$ bar using a stochastic velocity rescaling thermostat with damping frequency $t_v = 0.1$ ps and Parrinello-Rahman barostat with coupling frequency $t_p = 2.0$ ps, respectively. The Particle Mesh Ewald approach is used to compute long-range electrostatics with periodic boundary conditions applied to the x, y, and z directions. The conformational modes can be characterized by six free energy local minimas, which have been used in previous work (Lederer

---

[1] http://www.sgdml.org/
[2] https://archive.materialscloud.org/record/2022.45

(a)                                                        (b)

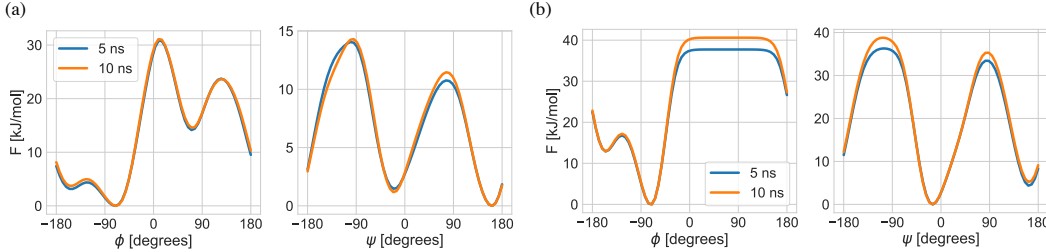

Figure 8: $F_\phi$ and $F_\psi$ have converged for the reference force field (a) and NequIP (b) at time 5 ns under Metadynamics.

et al., 2022). We initialize six simulations for each model from each of the six free energy local minimas.

**Implicit solvation.** The explicit solvent of 1164 water molecules is not the subject of study but adds a significant computational burden. In this task, we attempt to learn an implicit solvent model (ISM) of the alanine dipeptide, in which the explicit solvent environment is incorporated in the learned FF. The ISM is commonly used in drug design (Wang et al., 2019a) because it can speed up the computation by dramatically decreasing the number of particles required for simulation. In general, the mean-field estimation in ISM ignores the effect of solvent, thermal fluctuations, and solvent friction (Feig, 2007). Thus, molecular kinetics is not directly comparable to the explicit solvation simulation. However, the equilibrium configurations can be explicitly compared, as conducted in Chen et al. (2021).

**Metadynamics simulation.** Simulating energy barrier jump usually requires a long sampling of the trajectory in MD simulations. The conformational change of alanine dipeptide in water involves such a process, making it difficult to extract the complete free energy surface, i.e., the Ramachandran plot, in normal MD. In order to examine the learned ML FFs within a reasonable time limit, metadynamics (Laio & Parrinello, 2002) is employed to explore the learned FES of the solvated alanine dipeptide. Metadynamics is a widely used technique in atomistic simulations to accelerate the sampling of rare events and estimate the FES of a certain set of degrees of freedom. It is based on iteratively "filling" the potential energy using a sum of Gaussians deposited on a set of suitable collective variables (CVs) along the trajectory. At evaluation time, we perform metadynamics with dihedral angles $\phi$ and $\psi$ as CVs[3], starting from the configurations located at one of the six energy minimums in the free energy surface indicated in Figure 5 (c). The Gaussians with height $h = 1.2$ and sigma $\sigma = 0.35$ are deposited every 1 ps centered on $\psi$ and $\phi$. As shown in Figure 8, the estimated FES of both $\phi$ and $\psi$ do not significantly change after 5 ns. In addition, the height of the bias gaussian potential smoothly converges to $\sim 0$ in the time limit of 5 ns. Therefore, a simulation time of 5 ns is sufficient for the convergence of the metadynamics. This metadynamics simulation of alanine dipeptide with AMBER force-fields is carried out using GROMACS (Abraham et al., 2015) integrated with the PLUMED library (Tribello et al., 2014; Colón-Ramos et al., 2019) of version 2.8.

## C    EXPERIMENTAL DETAILS

The Open Catalyst Project[4] codebase and the official codebases of DeepPot-SE[5] and NequIP[6] are all publicly available. We build our MD simulation framework based on the Atomic Simulation Environment (ASE) library[7] (Larsen et al., 2017).

**Hyperparameters.** We adopt the original model hyperparameters in the respective papers and find they can produce good force prediction results that match the trend and numbers for MD17 reported in previous work. As we introduce new datasets, we set training hyperparameters such as the batch size and summarize them in Table 7. For water and LiPS, we use a batch size of 1 like in previous

---

[3]In practice, the selection of suitable collective variables can be a case-by-case challenge (Sidky et al., 2020).
[4]https://github.com/Open-Catalyst-Project/ocp
[5]https://github.com/deepmodeling/deepmd-kit
[6]https://github.com/mir-group/nequip
[7]https://gitlab.com/ase/ase

Table 7: Default training-related hyperparameters for each dataset. [*]We adopt the original batch size from respective papers when available for MD17. DeepPot-SE: 4; SchNet: 100; DimeNet: 32; GemNet-T/dT: 1; NequIP: 5. We use a batch size of 8 for ForceNet.

| Dataset | Training dataset size | Batch size | Max epoch | LR patience | Longest simulation time |
|---|---|---|---|---|---|
| MD17 | 9,500 | 1-100[*] | 2,000 | 5 epochs | 20 hours |
| Water-1k | 950 | 1 | 10,000 | 50 epochs | 28 hours |
| Water-10k | 9,500 | 1 | 2,000 | 5 epochs | 28 hours |
| Water-90k | 85,500 | 1 | 400 | 3 epochs | 28 hours |
| Alanine dipeptide | 38,000 | 5 | 2,000 | 5 epochs | 75 hours |
| LiPS | 19,000 | 1 | 2,000 | 5 epochs | 7 hours |

Table 8: Results on Water-1k. Force MAE is reported in the unit of [meV/Å]; Stability is reported in the unit of [ps]; Diffusivity MAE is reported in the unit of $[10^{-9} \mathrm{~m}^2/\mathrm{s}]$; RDF MAE and FPS are unitless.

| | DeepPot-SE | SchNet | DimeNet | ForceNet | GemNet-T | GemNet-dT | NequIP |
|---|---|---|---|---|---|---|---|
| Force | 6.7 | 13.1 | 3.5 | 13.6 | 5.0 | 29.2 | 1.4 |
| Stability | $108_{(117)}$ | $175_{(56)}$ | $4_{(4)}$ | $13_{(7)}$ | $6_{(7)}$ | $0_{(0)}$ | $500_{(0)}$ |
| $RDF_{(O,O)}$ | $0.17_{(0.10)}$ | $0.52_{(0.05)}$ | $0.46_{(0.22)}$ | $0.86_{(0.09)}$ | $0.62_{(0.48)}$ | - | $0.07_{(0.02)}$ |
| $RDF_{(H,H)}$ | $0.13_{(0.09)}$ | $0.24_{(0.02)}$ | $0.33_{(0.15)}$ | $0.56_{(0.04)}$ | $0.35_{(0.21)}$ | - | $0.07_{(0.02)}$ |
| $RDF_{(H,O)}$ | $0.28_{(0.15)}$ | $0.54_{(0.01)}$ | $0.43_{(0.17)}$ | $1.44_{(0.09)}$ | $0.71_{(0.65)}$ | - | $0.26_{(0.07)}$ |
| Diffusivity | 0.24 | 1.79 | - | - | - | - | 0.37 |

work (Batzner et al., 2022) as each structure already contains a reasonable number of atoms and interactions. Following previous work, we use an initial learning rate of 0.001 for all experiments except for NequIP, which uses 0.005 as the initial learning rate in the original paper. For models that minimize a mixture of force loss and energy loss, we set the force loss coefficient $\lambda_F$ to be 1000 and the energy loss coefficient $\lambda_E$ to be 1, if not specified in the original paper. A higher force loss coefficient is common in previous work (Zhang et al., 2018a; Batzner et al., 2022) as simulations do not directly rely on the energy.

Notably, NequIP proposed several sets of hyperparameters for different datasets, including MD17, a water+ice dataset from Zhang et al. 2018a, LiPS, etc. We follow the MD17 hyperparameters of NequIP for our MD17 and alanine dipeptide datasets; the water+ice hyperparameters of NequIP for our water dataset; and the LiPS hyperparameters of NequIP for the same LiPS dataset. For DeepPot-SE, we adopted hyperparameters introduced in Zhang et al. 2018a. The only architectural adjustment we made is because we observed training instability for ForceNet on water using the original hyperparameters. We resolve this issue by reducing the network width from 512 to 128 for ForceNet in our water experiments.

To facilitate benchmarking with a reasonable computational budget, we stop the training of a model if either of the following conditions is met: (1) a maximum training time of 7 days is reached on an NVIDIA Tesla V100-PCIe GPU; (2) a maximum number of epochs specified in Table 7 is reached; (3) The learning rate drops below $10^{-6}$ with a `ReduceLROnPlateau` scheduler with factor 0.8 and learning rate (LR) patience specified in Table 7. We also report the longest time for an ML model to finish our benchmark simulation in Table 7. All numbers are results of NequIP. The high computational cost for evaluating MD simulations has been a major consideration in designing our benchmark datasets and metrics. Training of DeepPot-SE is efficient and we follow the training setup specified in Zhang et al. 2018a.

**Complete water results.** We present results on water-1k in Table 8 and results on water-90k in Table 9. Results on water-10k is presented in Table 4 in the main text. All models generally achieve lower force error when trained with more data, but stability and estimation of ensemble statistics don't necessarily improve. In particular, DeepPot-SE shows clear improvement with more training data and becomes as good as NequIP on water-90k. SchNet demonstrates significant improvement in stability, but the estimation of ensemble statistics does not improve. This may be due to the limited accuracy of SchNet coming from the limited expressiveness of the invariant atomic representation.

Table 9: Results on Water-90k. Force MAE is reported in the unit of [meV/Å]; Stability is reported in the unit of [ps]; Diffusivity MAE is reported in the unit of $[10^{-9} \text{ m}^2/\text{s}]$; RDF MAE and FPS are unitless.

|  | DeepPot-SE | SchNet | DimeNet | ForceNet | GemNet-T | GemNet-dT | NequIP |
|---|---|---|---|---|---|---|---|
| Force | 5.9 | 8.4 | 1.7 | 8.6 | 0.7 | 1.1 | 1.4 |
| Stability | $500_{(0)}$ | $299_{(70)}$ | $36_{(9)}$ | $9_{(12)}$ | $20_{(9)}$ | $8_{(10)}$ | $500_{(0)}$ |
| $\text{RDF}_{(O,O)}$ | $0.07_{(0.02)}$ | $0.67_{(0.03)}$ | $0.21_{(0.03)}$ | $1.31_{(0.49)}$ | $0.35_{(0.23)}$ | $0.20_{(0.01)}$ | $0.06_{(0.01)}$ |
| $\text{RDF}_{(H,H)}$ | $0.05_{(0.01)}$ | $0.31_{(0.02)}$ | $0.14_{(0.01)}$ | $0.82_{(0.26)}$ | $0.25_{(0.19)}$ | $0.16_{(0.01)}$ | $0.04_{(0.01)}$ |
| $\text{RDF}_{(H,O)}$ | $0.29_{(0.08)}$ | $0.67_{(0.04)}$ | $0.18_{(0.02)}$ | $2.05_{(0.60)}$ | $0.24_{(0.06)}$ | $0.26_{(0.02)}$ | $0.25_{(0.06)}$ |
| Diffusivity | 0.35 | 1.97 | - | - | - | - | 0.18 |

Table 10: Water-1k results on NequIP with various model sizes and radius cutoffs.

|  | Force | Stability | $\text{RDF}_{(O,O)}$ | $\text{RDF}_{(H,H)}$ | $\text{RDF}_{(H,O)}$ | Diffusivity | FPS |
|---|---|---|---|---|---|---|---|
| Width=64, r=4 | 3.5 | $500_{(0)}$ | $0.07_{(0.02)}$ | $0.05_{(0.01)}$ | $0.27_{(0.06)}$ | 0.38 | 8.2 |
| Width=32, r=6 | 1.5 | $500_{(0)}$ | $0.06_{(0.01)}$ | $0.05_{(0.01)}$ | $0.26_{(0.06)}$ | 0.25 | 5.2 |
| Width=64, r=5 | 1.6 | $500_{(0)}$ | $0.07_{(0.02)}$ | $0.05_{(0.01)}$ | $0.27_{(0.06)}$ | 0.31 | 4.9 |
| Width=64, r=6 | 1.4 | $500_{(0)}$ | $0.07_{(0.02)}$ | $0.07_{(0.02)}$ | $0.26_{(0.07)}$ | 0.37 | 3.9 |
| Width=128, r=6 | 1.5 | $500_{(0)}$ | $0.07_{(0.02)}$ | $0.05_{(0.01)}$ | $0.29_{(0.07)}$ | 0.37 | 2.5 |

**Influence of model size.** Table 10 shows an ablation study over the model size and radius cutoff of NequIP over water-1k. We observe that all models are highly stable and attain equally good performance in simulation-based metrics. Although a small radius cutoff of 4 leads to worse performance in force prediction, it is more computationally efficient and preserves the trajectory statistics. These results show that there exists a trade-off between accuracy and efficiency when choosing the hyperparameters of an ML force field, and force error may not be the preferred criterion for model selection.

**Large water system of 512 molecules.** To study model performance in generalizing to a larger system and model scalability, we evaluate models trained on the water-10k dataset on a dataset with 512 water molecules simulated for 1 ns, using the same reference force field. Given the high cost of simulating a large system, we simulate 5 trajectories of 150 ps long for each model. The results are shown in Table 11. We observe that all models suffer slightly higher force errors compared to evaluation over the 64-molecule water system. In terms of stability, NequIP and SphereNet always remains stable for the entire 150 ps. However, SphereNet does not produce correct ensemble properties. SchNet is the third stable model, while all other models are not stable enough for diffusivity computation. DimeNet, GemNet-T, and GemNet-dT are not stable through the entire simulation, but can produce decent RDF results. Noticeably, the stability of DeepPot-SE drops significantly. We hypothesize that the lack of message passing limits its capability in capturing long-range interactions and thus limits the performance in a larger system.

**Stability's relation with dataset size.** We extract the force and stability results for each model from Figure 4 to make Figure 9 to better illustrate the relation between stability and dataset size for each model. We observe that while more data almost always reduce force error, stability does not necessarily improve. In particular, NequIP is highly stable across all dataset size. DeepPot-SE and SchNet has significant improvement in stability with more data. While for DimeNet ForceNet, and GemNet, more training data does not bring significant stability improvement. Section 6 contains detailed discussions on the causes of instability and potential solutions to improve stability.

**Stability's relation with training epochs.** We study the evolution of simulation stability in the training process of an ML force field. We take the SchNet model on the MD17 molecule salicylic acid, and save the checkpoint at 100, 200, 300, 400, and 500 epochs. We conduct 5 simulations of 300 ps using each checkpoint. Figure 10 shows the force error and stability of the model at different stages of training. We observe that the force error decreases as training progress, and the stability improves to be stable across the entire 300 ps simulation and training epoch 300. This result reveals that thorough training is important to both accuracy and stability of ML force fields.

Table 11: Results on the large water system with 512 molecules, with models trained on the water-10k dataset (64-molecule water system). *SphereNet on Water-large requires more memory than Tesla V100 supports. We run its simulations on faster NVIDIA A100 cards so the FPS is not entirely comparable to other models.

| | DeepPot-SE | SchNet | DimeNet | PaiNN | SphereNet* | ForceNet | GemNet-T | GemNet-dT | NequIP |
|---|---|---|---|---|---|---|---|---|---|
| Force | 10.6 | 12.1 | 5.1 | 9.7 | 18.4 | 13.2 | 5.6 | 4.2 | 7.7 |
| Stability | $19_{(22)}$ | $118_{(58)}$ | $38_{(13)}$ | $16_{(12)}$ | $150_{(0)}$ | $8_{(0)}$ | $45_{(25)}$ | $50_{(9)}$ | $150_{(0)}$ |
| $RDF_{(O,O)}$ | $0.23_{(0.06)}$ | $0.62_{(0.01)}$ | $0.17_{(0.03)}$ | $0.31_{(0.06)}$ | $0.93_{(0.02)}$ | $0.74_{(0.02)}$ | $0.22_{(0.16)}$ | $0.16_{(0.02)}$ | $0.10_{(0.01)}$ |
| $RDF_{(H,H)}$ | $0.24_{(0.06)}$ | $0.30_{(0.04)}$ | $0.12_{(0.03)}$ | $0.21_{(0.05)}$ | $0.42_{(0.01)}$ | $0.51_{(0.02)}$ | $0.15_{(0.11)}$ | $0.11_{(0.01)}$ | $0.07_{(0.00)}$ |
| $RDF_{(H,O)}$ | $0.67_{(0.27)}$ | $0.55_{(0.01)}$ | $0.17_{(0.02)}$ | $0.29_{(0.05)}$ | $0.97_{(0.03)}$ | $1.38_{(0.05)}$ | $0.23_{(0.12)}$ | $0.16_{(0.02)}$ | $0.12_{(0.02)}$ |
| Diffusivity | - | 2.54 | - | - | 2.98 | - | - | - | 0.89 |
| FPS | 80.7 | 23.1 | 3.5 | 17.4 | 0.8 | 11.9 | 2.2 | 5.3 | 0.7 |

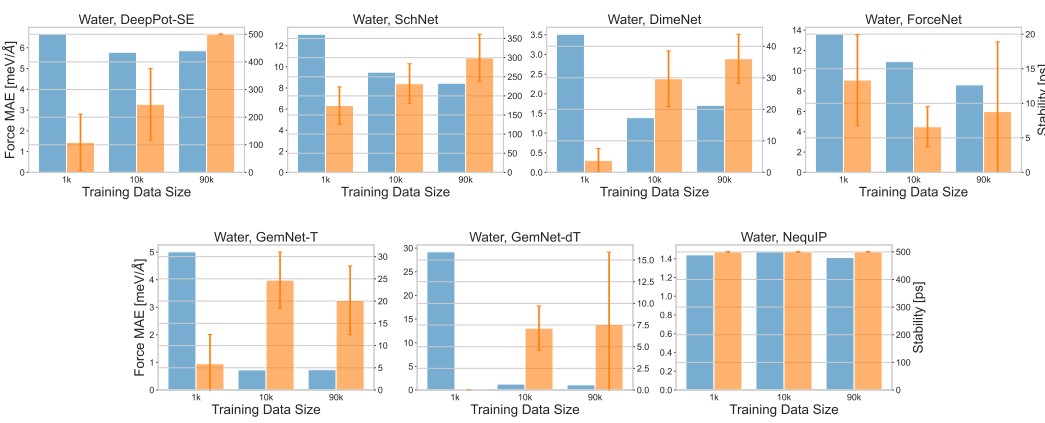

Figure 9: The force error and stability of all models on the water dataset, with varying training dataset size: 1k, 10k, and 90k.

**Distribution of interatomic distances for MD17.** Figure 11 shows the $h(r)$ curves for all models and molecules benchmarked. We randomly selected one simulation out of the five simulations we conducted for each model and molecule. We observe that due to lack of stability, DeepPot-SE produces noisy $h(r)$ on Aspirin. ForceNet does not manage to learn the correct interatomic interactions and produces incorrect $h(r)$ curves. Most models are able to produce $h(r)$ that match well with the reference, with SchNet being less accurate on `Aspirin` and `Ethanol`.

**RDFs for water.** Selected RDF curves for water-1k/10k/90k are in Figure 12, Figure 13, and Figure 14. Most noisy curves are due to insufficient sampling time, which results in a small number of frames to be averaged in obtaining the RDF curves. We observe that SchNet and ForceNet produce inaccurate curves that are not very noisy, showing that their failure is not entirely due to lack of stability but because of inaccurate modeling of interactions caused by limited expressiveness and lower sample efficiency. Further, we note that the reference curves have zero values below a certain threshold, as any pair of atoms cannot get too close to each other. However, DimeNet and GemNet-T exhibit abnormal high values for very small distances, indicating the simulations have gone into nonphysical configurations and collapsed.

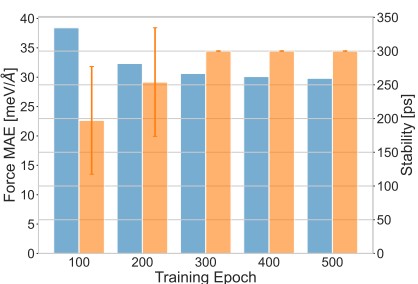

Figure 10: The force error and stability of SchNet for simulating Salicylic acid, as traing progress.

**RDFs for LiPS.** As shown in Figure 15, DeepPot-SE does not manage to stay stable on LiPS. ForceNet learns inaccurate interactions and produces inaccurate RDFs. All other models can produce highly accurate RDF and can reproduce Li-ion diffusivity relatively accurately, as demonstrated in Table 6.

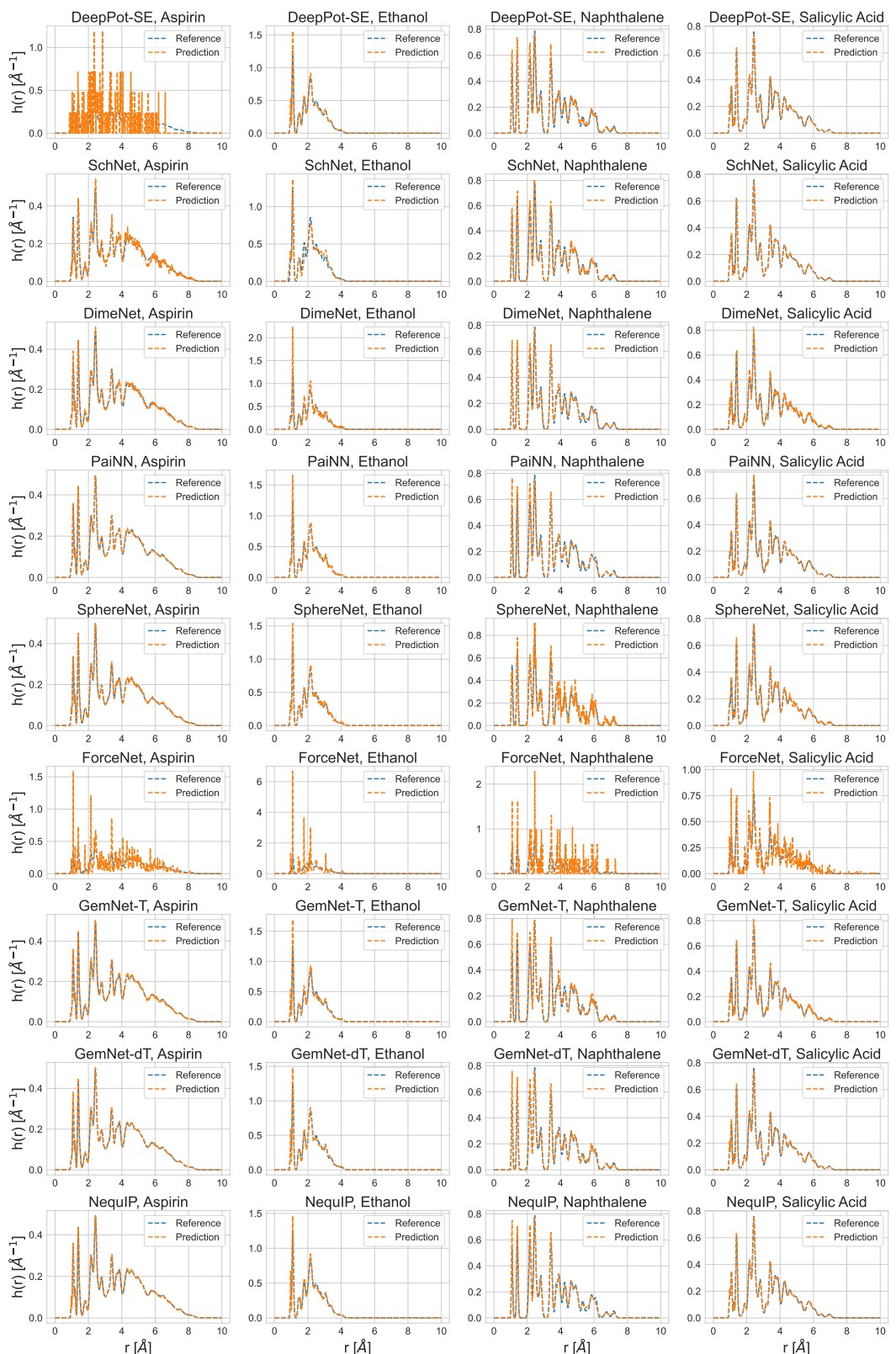

Figure 11: h(r) of md17.

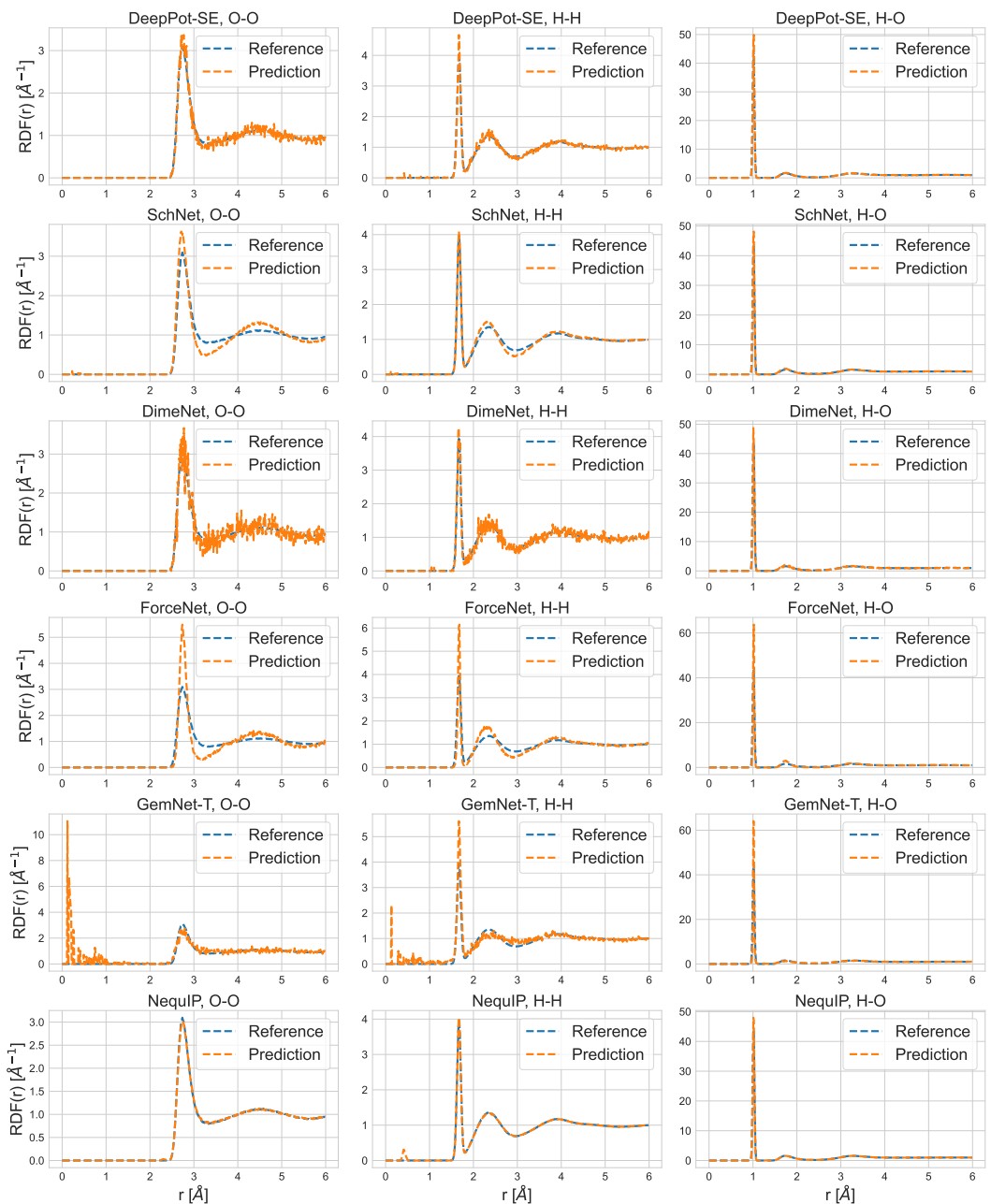

Figure 12: RDFs of Water-1k. GemNet-dT does not remain stable for more than 1 ps and is therefore not feasible for RDF computation.

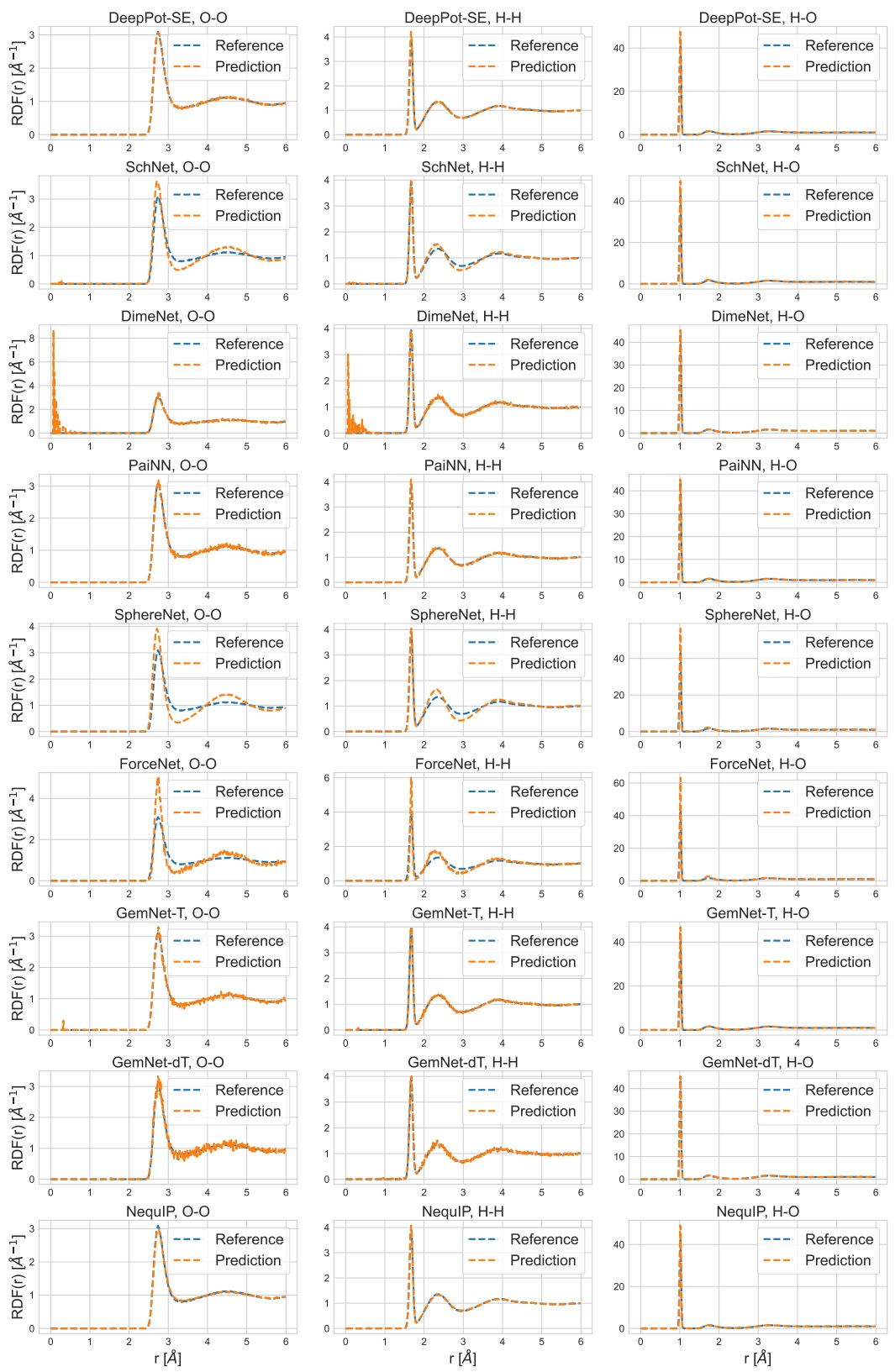

Figure 13: RDFs of Water-10k.

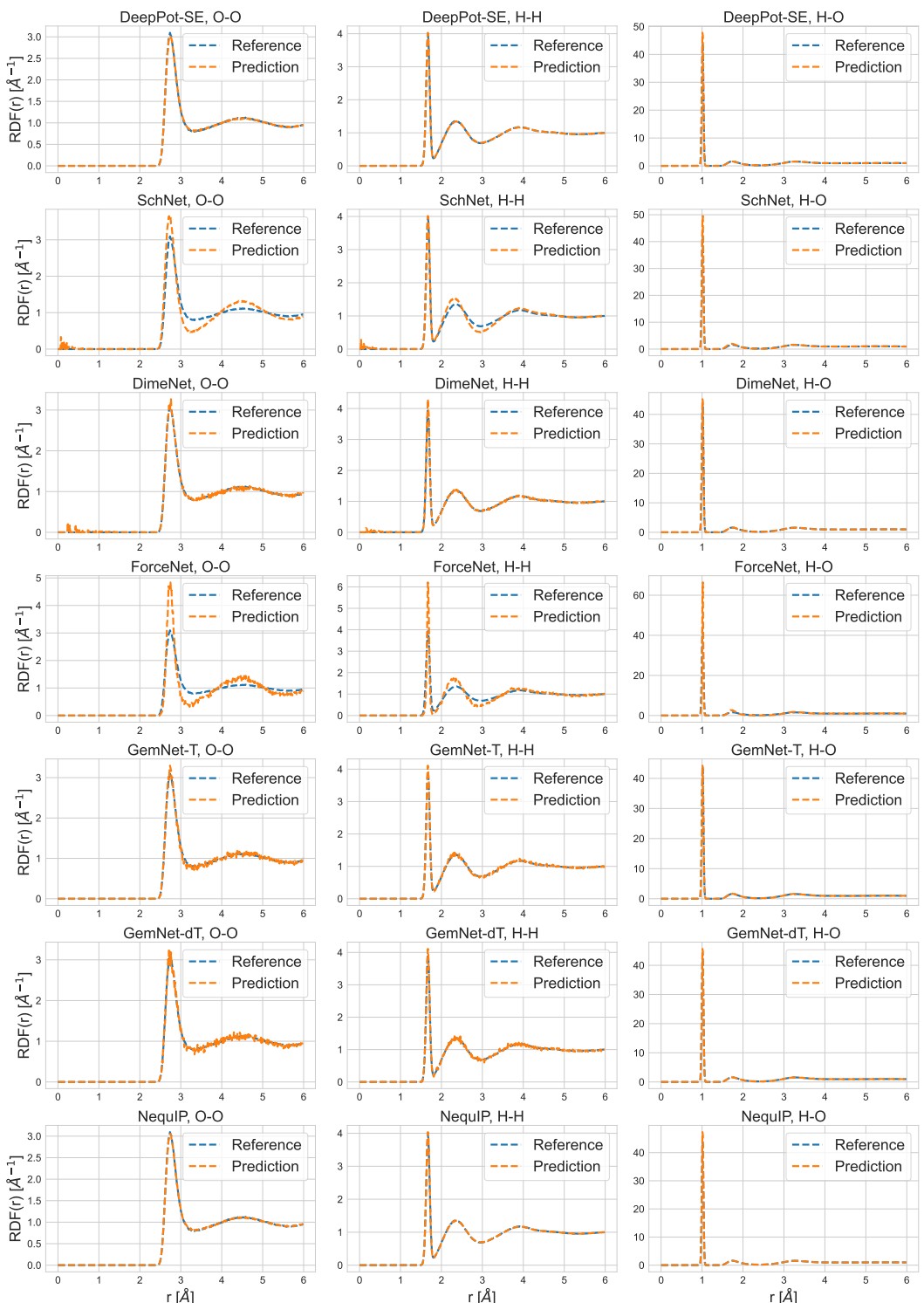

Figure 14: RDFs of Water-90k.

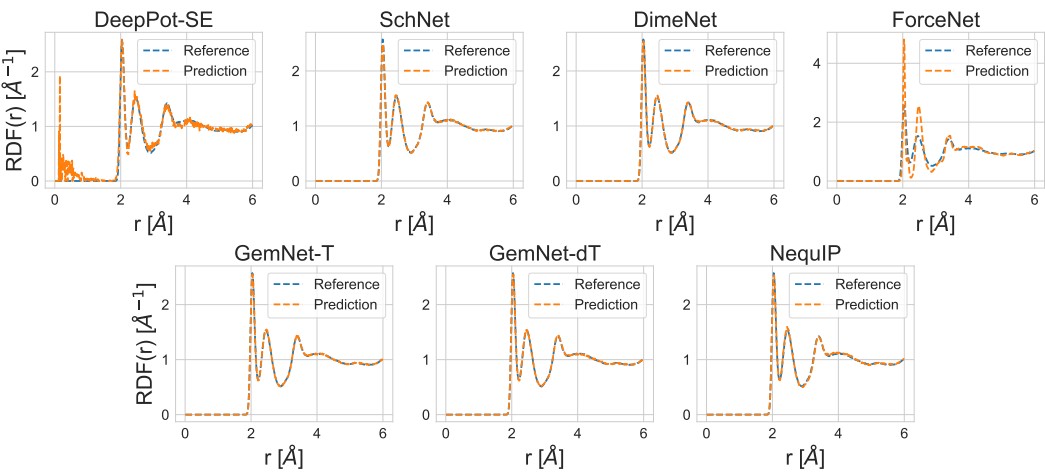

Figure 15: RDFs of LiPS.

