# OpenReview forum: "Forces are not Enough: Benchmark and Critical Evaluation for Machine Learning Force Fields with Molecular Simulations"
_ICLR.cc/2023/Conference — Submitted to ICLR 2023_

### Official Review · Reviewer_t1Na · 2022-10-23

**Confidence:** 3
**Correctness:** 4
**Technical Novelty And Significance:** 2
**Empirical Novelty And Significance:** 4
**Recommendation:** 6

**Clarity, Quality, Novelty And Reproducibility:**

I think the paper is written clearly and supplementary materials provide code to reproduce result. However, the novelty seems to be rather limited except several stability metrics for the ML-FF.

**Strength And Weaknesses:**

Strength:
1. The results of the paper is very convincing given the fact that it utilizes more training configurations than existing work such as GemNet.
2. The illustration of this paper is extraordinary such as Figure 5 and Figure 6.

Weakness & Questions:
1. One of my major concern is the paper seems more like a benchmarking paper. All of the models and datasets used in the paper already exist and the instability of force field simulation has been discussed by previous work.
2. I notice the result on MD17 is different from references for method like GemNet, I notice it's mainly because you are using more training data and in their paper GemNet is better than NequIP.
3. How is the stability related with training epoch and amount of training samples?

**Summary Of The Paper:**

This paper focus on examining the current state of force field machine learning model in molecule dynamics. Specifically, it proposes several simulation based metrics to evaluate the stability of existing models. For the SoTA model, it provides further analysis on simulation failures and reveals underlying reasons.

**Summary Of The Review:**

Based on my understanding, this work states the challenge of force field simulation stability, which could possibly inspire more research effort on this topic. The relation of stability and model training is not justified in the current version and novelty of the paper is somehow limited.

---

> ### Author Response · Authors · 2022-11-11
> **Response part 1**
>
> We thank reviewer t1Na for constructive comments and suggestions. We address each of the reviewer’s concerns below.
>
> > One of my major concern is the paper seems more like a benchmarking paper. All of the models and datasets used in the paper already exist and the instability of force field simulation has been discussed by previous work. Novelty of the paper is somehow limited.
>
> Although the datasets and models are from existing work, we argue our benchmark, experimental analysis, and insights are **novel and significant contributions to the ML community.** We believe to convincingly evaluate the capability of ML models for MD simulations, we need to use well-known MD systems and observables with scientific grounding. On the other hand, we design the benchmark metrics and stability criterions, along with the simulation protocols to facilitate meaningful benchmarking with reasonable computational requirements. Our results are obtained from simulations exceeding 3000 GPU hours, and reveal novel insights into the growing field of ML for MD simulations. We believe our benchmark is an important step in pushing the field forward.
>
> As stability has been discussed in recent work [1,2], we want to further clarify the novel contribution of this work compared to related efforts. In particular, to our knowledge, our work is the first to create a complete and diverse MD benchmark for ML with (1) actual simulations, (2) novel and scientifically grounded metrics, and (3) many baseline models.
>
> - We include diverse and representative systems to thoroughly evaluate model performance in different MD applications. Existing works [1,2] focus on a single model at one type of system.
> - On top of stability, we propose observable-based metrics for each system to evaluate model accuracy. A stable but inaccurate model is still not useful for practical MD applications.
> - As the first benchmark across different models, we reveal novel insights into ML for MD simulations. For example, by comparing different models we find a model can have higher force error but is more stable, so model development should not focus on minimizing the force error. Such observation is independent of approaches such as extending data [1] for improving stability.
> - In addition to stability, we elucidate many other considerations in using ML models for MD simulations, such as efficiency, energy conservation, mode coverage, and failure modes.
>
> As ICLR does **NOT** have a benchmark track, many novel benchmark papers [3,4,5,6,7] have been accepted to the ICLR conference in past years. Benchmarks have been a significant component in the development of machine learning research, especially in bridging the gap between ML and the respective fields of application.
>
> > I notice the result on MD17 is different from references for method like GemNet, I notice it's mainly because you are using more training data and in their paper GemNet is better than NequIP.
>
> We adopt a larger training data size as the simulation task is a lot more challenging than force prediction. To accommodate non-equivariant models which tend to be less sample-efficient, we incorporate a moderate data budget. Our reported force errors are generally better than the respective papers.
>
> To clarify the comparison between NequIP and GemNet: the GemNet paper [8] uses early results of the NequIP pre-print in the paper, which shows GemNet outperforms NequIP. NequIP was later updated and published, with improved results [9]. In the latest version of NequIP, it is shown to outperform GemNet on MD17 with a training budget of 1,000 samples.

---

> > ### Author Response · Authors · 2022-11-11
> > **Response part 2**
> >
> > > How is the stability related with training epoch and amount of training samples?
> >
> > Thank you for the constructive questions. The stability’s relation with the amount of training samples is demonstrated in Figure 4. As more training samples almost always improve force prediction, their relation with the stability is not consistent across different models. In our experiments, NequIP is stable across three dataset sizes. DeepPot-SE and SchNet achieve significant improvement in stability with more training data, while DimeNet, ForceNet, and GemNet do not have significant stability improvement with more data.
> >
> > To better demonstrate this relation for each model, **we add a paragraph of discussion and Figure 9 to Appendix C where we show the trend of stability across dataset size for each model.** We also summarize this result in the table below. One standard deviation for applicable metrics is in subscript.
> >
> > | Model      | ('Force', '1k')   | ('Force', '10k')   | ('Force', '90k')   | ('Stability', '1k')   | ('Stability', '10k')   | ('Stability', '90k')   |
> > |:-----------|:-------------------|:--------------------|:--------------------|:-----------------------|:------------------------|:------------------------|
> > | DeepPot-SE | $6.7$              | $5.8$               | $5.9$               | $108_{(117)}$          | $247_{(147)}$           | $500_{(0)}$             |
> > | DimeNet    | $3.5$              | $1.4$               | $1.7$               | $4_{(4)}$              | $30_{(10)}$             | $36_{(9)}$              |
> > | ForceNet   | $13.6$             | $10.9$              | $8.6$               | $13_{(7)}$             | $7_{(3)}$               | $9_{(12)}$              |
> > | GemNet-T   | $5.0$              | $0.7$               | $0.7$               | $6_{(7)}$              | $25_{(7)}$              | $20_{(9)}$              |
> > | GemNet-dT  | $29.2$             | $1.3$               | $1.1$               | $0_{(0)}$              | $7_{(3)}$               | $8_{(10)}$              |
> > | NequIP     | $1.4$              | $1.5$               | $1.4$               | $500_{(0)}$            | $500_{(0)}$             | $500_{(0)}$             |
> > | SchNet     | $13.1$             | $9.5$               | $8.4$               | $175_{(56)}$           | $232_{(59)}$            | $299_{(70)}$            |
> >
> > To better understand the relation between stability and training epochs, **we conduct new experiments** to study this relation using the SchNet model on the MD17 molecule Salicylic acid. We save model checkpoints every 100 epochs and simulate 5 trajectories from 5 random initial states sampled from the test set. We find longer training improves both force error and stability, as stability converges at the late stage of training. Therefore, thorough training is important to both the accuracy and stability of ML force fields. **We add a paragraph of discussion and Figure 10 to Appendix C where we demonstrate the experiment results.** We also summarize this result in the table below. One standard deviation for applicable metrics is in subscript.
> >
> > | Training Epoch | 100| 200 | 300 | 400 | 500 |
> > | -- | -- | -- | -- | -- | -- |
> > | Force MAE [meV/A] | 38.4| 32.3| 30.6| 30.1| 29.8|
> > | Stability [ps] | $197.3_{(91.2)}$ | $254.1_{(91.8)}$ | $300_{(0)}$ | $300_{(0)}$ | $300_{(0)}$ |
> >
> > We look forward to further discussion if you have additional feedback.

---

> > > ### Author Response · Authors · 2022-11-11
> > > **Response part 3**
> > >
> > > Reference:
> > >
> > > [1] Stocker, Sina, et al. "How Robust are Modern Graph Neural Network Potentials in Long and Hot Molecular Dynamics Simulations?." Machine Learning: Science and Technology (2022).
> > >
> > > [2] Zhai, Yaoguang, et al. "A “short blanket” dilemma for a state-of-the-art neural network potential for water: Reproducing properties or learning the underlying physics?." (2022).
> > >
> > > [3] Fu, Xiang, et al. "Simulate Time-integrated Coarse-grained Molecular Dynamics with Geometric Machine Learning." arXiv preprint arXiv:2204.10348 (2022).
> > >
> > > [4] Santurkar, Shibani, Dimitris Tsipras, and Aleksander Madry. "Breeds: Benchmarks for subpopulation shift." International Conference on Learning Representations. 2021.
> > >
> > > [5] Li, Wenda, et al. "Isarstep: a benchmark for high-level mathematical reasoning." International Conference on Learning Representations. 2021.
> > >
> > > [6] Mehrjou, Arash, et al. "GeneDisco: A Benchmark for Experimental Design in Drug Discovery." International Conference on Learning Representations. 2022.
> > >
> > > [7] Wei, Jiaheng, et al. "Learning with noisy labels revisited: A study using real-world human annotations." International Conference on Learning Representations. 2022.
> > >
> > > [8] Gasteiger, Johannes, Florian Becker, and Stephan Günnemann. "Gemnet: Universal directional graph neural networks for molecules." Advances in Neural Information Processing Systems 34 (2021): 6790-6802.
> > >
> > > [9] ​​Batzner, Simon, et al. "E (3)-equivariant graph neural networks for data-efficient and accurate interatomic potentials." Nature communications 13.1 (2022): 1-11.

---

### Official Review · Reviewer_CmqU · 2022-10-24

**Confidence:** 3
**Correctness:** 4
**Technical Novelty And Significance:** 2
**Empirical Novelty And Significance:** 2
**Recommendation:** 6

**Clarity, Quality, Novelty And Reproducibility:**

Clarity:
The paper is well-organized and well-written.

Quality:
The claims are well-supported by the experiments.

Novelty:
Although this is not the first paper addressing the problem, no standard benchmark set is not available in the community before.

Reproducibility:
There is room for improvement in the description of evaluation procedures above.


Minor comments:
Page 15:  h(r) should be D in the Diffusivity coefficient paragraph.


**Details Of Ethics Concerns:**

I don't find any issue.

**Strength And Weaknesses:**

Strength
* Although it has been known that force accuracy is not enough metric for ML FFs in the community, the standardization of MD-based evaluation procedure is valuable for developing ML FFs.
* State-of-the-art ML FF models are evaluated in the experiments.
* Section 6 is a good introduction to the issues in the current ML FFs in ML community.

Weaknesses
* Reason for choosing the applications of MD simulations is not provided.  There is other popular applications, such as viscosity (Zhang et al., 2015) or thermal conductivity (Müller-Plathe, 1997).
* The procedural details of computing physical properties are not provided.   Since one of the promising directions for stabilizing dynamics is an extension of training data (Stocker et al., 2022; Takamoto et al., 2022), extended training datasets will be used in future work.  Therefore, standardizing the evaluation procedures is more important than fixing the dataset set.

Questions:
* Why did you perform six simulations only for Alanine dipeptide and five for others?   How did you find the number of simulations?
* How is "the stable part of the simulation" determined?

- Zhang et al.  "Reliable Viscosity Calculation from Equilibrium Molecular Dynamics Simulations: A Time Decomposition Method".  Journal of Chemical Theory and Computation (2015), 3537-3546.
- Müller-Plathe, Florian. "A simple nonequilibrium molecular dynamics method for calculating the thermal conductivity." The Journal of chemical physics (1997), 6082-6085.
- Takamoto et al., "Towards universal neural network potential for material discovery applicable to arbitrary combination of 45 elements".  Nature Communications (2022).

**Summary Of The Paper:**

The paper addresses the evaluation of force fields (FF) for atomistic simulations.
The authors propose benchmark datasets and metrics based on molecular dynamics(MD) simulation, a major use case of FFs.
When state-of-the-art machine learning(ML)-based FFs are evaluated on the benchmark datasets, conventional force accuracy is not always related to the performance of MD simulation.
The authors argue that there is room for improvement in the stability of ML-based FFs.

**Summary Of The Review:**

Although this is not the first paper addressing the problem, I would like to appreciate someone's contribution to standardizing the benchmarking of ML FFs.
I hope a more detailed description of the evaluation procedures is provided in the final version.

---

> ### Author Response · Authors · 2022-11-11
> **Response part 1**
>
> We thank reviewer CmqU for constructive comments and suggestions. We address each of the reviewer’s concerns below.
>
> > Reason for choosing the applications of MD simulations is not provided. There is other popular applications, such as viscosity (Zhang et al., 2015) or thermal conductivity (Müller-Plathe, 1997).
>
> We have three major considerations in choosing the MD systems: (1) diverse and representative of MD applications; (2) reasonable compute requirement; (3) meaningful and convincing to the scientific community. The benchmarked datasets span small molecules, water, peptide, and materials, and a variety of observables, including interatomic distances, RDF, diffusivity, and free energy surface reconstruction. We have added more details on the chosen datasets to the appendix.
>
> The metrics we chose (e.g., radial distribution function and diffusion coefficients) are commonly used to validate the developed molecular model in terms of structure and dynamics [1]. Viscosity and thermal conductivity are applications of MD that require significant domain knowledge to study, and they are not used for the development of classical/ML force fields. Moreover, the calculation of viscosity and thermal conductivity is much more time-consuming, which requires (1) larger system sizes (>1000) and (2) longer simulation time. The computation requirement makes them not suitable for benchmarks within reasonable time/compute.
>
> > The procedural details of computing physical properties are not provided.
>
> Thank you for the constructive feedback. We have added **a new section in Appendix A** to explain the benchmark procedures in more detail. We have also added a Jupyter Notebook named **“observable.ipynb”** containing the complete procedures for computing the metrics reported in the paper to the supplementary materials.
>
>
> > Why did you perform six simulations only for Alanine dipeptide and five for others? How did you find the number of simulations?
>
> We conduct six simulations for Alanine dipeptide because the free energy surface of alanine dipeptide has six local minima, which characterize different conformation modes. We initialize six simulations from these six local free energy minima in order to cover the diverse conformation of alanine dipeptide. The free energy surface with respect to the dihedral angles $\phi$ and $\psi$ are visualized in Figure 5(a), and the local minima are marked in Figure 5(c). These local energy minima are well-known for the alanine dipeptide system and have been used in previous work [2]. The explanation and reference are added to Appendix B.
>
> The other systems (MD17, water, and LiPS) don’t have a clear separation of conformation modes in their free energy surfaces. We, therefore, randomly sample five configurations from the test set to initialize simulations. We conduct five simulations to get a more accurate estimation of the metrics and error bars. Ideally, the simulations are the more, the better, and the longer, the better. However, due to the high cost of simulating MD, we choose five simulations per model to get a satisfying result with reasonable compute. For example, NequIP requires 28 hours on a Tesla V100 GPU to finish one simulation for the water system and 75 hours for one simulation for the alanine dipeptide system. The total GPU hours used in the paper’s experiments (not including training the models) exceeds 3,000 GPU hours.
>
> > How is "the stable part of the simulation" determined?
>
> "the stable part of the simulation" is the simulation trajectory before the first “instability” occurred. We propose stability criteria based on the equilibrium observable of each system (RDF for condensed phased systems and bond lengths for flexible molecules), which are described in detail in Appendix A. For example, for a flexible molecule, a state is “unstable” when there exists a bond whose length deviates from the average ground truth bond length by more than 0.5 Angstrom. The initial state is sampled from the test set, so is certainly “stable”. "The stable part of the simulation" would be the entire simulated trajectory before the first incidence of “any bond length deviates from ground truth by more than 0.5 Angstrom”.
>
> By using only the stable part for observable computation, we can still get meaningful metrics for quantities such as RDF, diffusivity, etc., even when the simulation collapses in the end. The unstable parts of the simulation are in unphysical configurations and cannot be used for observable computation.

---

> > ### Author Response · Authors · 2022-11-11
> > **Response part 2**
> >
> > > Although this is not the first paper addressing the problem, I would like to appreciate someone's contribution to standardizing the benchmarking of ML FFs.
> >
> > We thank the reviewer for appreciating our contribution. We want to further clarify the novel contribution of this work compare to related recent efforts [3,4]:
> >
> > - We include diverse and representative systems to thoroughly evaluate model performance in different MD applications. Existing works [3,4] focus on a single model at one type of system.
> > - On top of stability, we propose observable-based metrics for each system to evaluate model accuracy. A stable but inaccurate model is still not useful for practical MD applications.
> > - As the first benchmark across different models, we reveal novel insights into ML for MD simulations. For example, by comparing different models we find a model can have higher force error but is more stable, so model development should not focus on minimizing the force error. Such observation is independent of approaches such as extending data [3] for improving stability.
> > - In addition to stability, we elucidate many other considerations in using ML models for MD simulations, such as efficiency, energy conservation, mode coverage, and failure modes.
> >
> > We argue our benchmark, experimental analysis, and insights are **novel and significant contributions to the ML community.** Machine learning for MD simulation is a fast-growing field, but the force error-based benchmarks cannot reflect the practical utilization. We carefully curate representative systems of practical significance and reasonable compute requirements, and establish metrics with scientific grounding. We conduct thorough experiments to reveal the challenges and novel insights into ML for MD simulations, and we believe our benchmark is an important step in pushing the field forward.
> >
> > We look forward to further discussion if you have additional feedback.
> >
> >
> > Reference:
> >
> > [1] Henderson, R. L. "A uniqueness theorem for fluid pair correlation functions." Physics Letters A 49.3 (1974): 197-198.
> >
> > [2] Lederer, Jonas, et al. "Automatic Identification of Chemical Moieties." arXiv preprint arXiv:2203.16205 (2022).
> >
> > [3] Stocker, Sina, et al. "How Robust are Modern Graph Neural Network Potentials in Long and Hot Molecular Dynamics Simulations?." Machine Learning: Science and Technology (2022).
> >
> > [4] Zhai, Yaoguang, et al. "A “short blanket” dilemma for a state-of-the-art neural network potential for water: Reproducing properties or learning the underlying physics?." (2022).

---

### Official Review · Reviewer_j9ZS · 2022-10-24

**Confidence:** 4
**Correctness:** 4
**Technical Novelty And Significance:** 2
**Empirical Novelty And Significance:** 3
**Recommendation:** 6

**Clarity, Quality, Novelty And Reproducibility:**

Clarity: good

Quality: ok

Novelty: The proposed evaluation metrics and analysis contribute to most of the novelty. Datasets are all collected from existing ML MD works, and no novel method is proposed in this work.

Reproducibility: should be good


**Strength And Weaknesses:**

**Strength**

1. This benchmark provides comprehensive datasets for four different MD systems, including small molecules, liquid water, peptide, and materials, which is convenient for users who are interested in MD simulation. Moreover, the provided open-source codebase for training and simulation would benefit future works.

2. This work proposes a series of evaluation metrics for MD simulation, which haven’t been well studied in existing works. The authors conduct a thorough analysis with SOTA methods and show that the widely used force prediction is not sufficient for the evaluation and suggest stability should be a key metric to improve. I believe the proposed metrics are very meaningful, and these observations reveal insights for MD simulation.

**Weaknesses**

1. Missing related works. For example, SphereNet[1] is a recent work that has good performance for PES prediction, and TorchMD[2] is a recent DL framework for MD simulation. The authors need to discuss missing related works in Section 2 and add experimental comparison.

2. More data are encouraged to include in this benchmark. For example, include all eight molecules from MD17. In addition, chignolin used in [2] is a peptide that is extensively studied with MD. It can be added to your peptide track. Moreover, polymers used in [3] could also be added to your material track.

 3. Since this is a benchmark work without proposing any novel technical method, I would suggest submitting the paper to a benchmark track rather than ICLR main track.

Reference:

[1]. Liu, Yi, et al. "Spherical message passing for 3d molecular graphs." International Conference on Learning Representations. 2021.

[2]. Doerr, Stefan, et al. "Torchmd: A deep learning framework for molecular simulations." Journal of chemical theory and computation 17.4 (2021): 2355-2363.

[3]. Fu, Xiang, et al. "Simulate Time-integrated Coarse-grained Molecular Dynamics with Geometric Machine Learning." arXiv preprint arXiv:2204.10348 (2022).


**Summary Of The Paper:**

This work proposes a novel benchmark for machine learning-based molecular dynamic simulations. The proposed benchmark includes four datasets and several quantitative evaluation metrics. Experiments and analysis with a collection of SOTA ML models are performed, and an open-source codebase is provided.

**Summary Of The Review:**

The proposed evaluation metrics are meaningful, the analysis of SOTA methods is insightful, and the open-source codebase can benefit future works. However, several important related works are missing. And I’m concerned about the novelty. Therefore, I’m inclined to reject.

---

> ### Author Response · Authors · 2022-11-11
> **Response part 1**
>
> We thank reviewer j9ZS for helpful questions and suggestions. We address each of the reviewer’s concerns below.
>
> > Missing related works. For example, SphereNet[1] is a recent work that has good performance for PES prediction, and TorchMD[2] is a recent DL framework for MD simulation. The authors need to discuss missing related works in Section 2 and add experimental comparison.
>
> Thank you for pointing out the related works. We have added references to SphereNet [1], and TorchMD [2] is already referenced in the paper. We note that TorchMD is a framework for implementing ML force fields. It is not a model by itself. According to the original paper [2], a SchNet-based model is being implemented with the TorchMD framework and used for the experiments in [2]. Our implementation uses PyTorch for the models and ASE [3] for doing MD simulations. TorchMD can be a great alternative to build a codebase with.
>
> We are running experiments to add SphereNet and PaiNN [4] to baseline comparisons. Note that running MD simulations requires significant computation. With limited compute resources, we are only able to finish the experiments for PaiNN on MD17 and water and a large water system (512 molecules) at this point. We are currently running SphereNet experiments and will add the results when they become available.
>
> **We have added PaiNN to Figure1, Figure 3, Table 2, Table 3, and Table 4.** We will update more results as they become available. We also summarize the results in the table below. PaiNN attains decent performance on MD17 molecules but is not stable for water. One standard deviation for applicable metrics is in subscript. The computation of diffusivity requires 100 ps stable simulation.
>
> PaiNN on MD17:
>
> |                                  | PaiNN           |
> |:---------------------------------|:----------------|
> | ('Aspirin', 'Force')            | $9.2$           |
> | ('Aspirin', 'Stability')        | $159_{(121)}$   |
> | ('Aspirin', 'h(r)')              | $0.04_{(0.01)}$ |
> | ('Aspirin', 'FPS')              | $85.8$          |
> | ('Ethanol', 'Force')            | $5.0$           |
> | ('Ethanol', 'Stability')        | $86_{(109)}$    |
> | ('Ethanol', 'h(r)')              | $0.15_{(0.08)}$ |
> | ('Ethanol', 'FPS')              | $87.3$          |
> | ('Naphthalene', 'Force')        | $3.8$           |
> | ('Naphthalene', 'Stability')    | $300_{(0)}$     |
> | ('Naphthalene', 'h(r)')          | $0.13_{(0.00)}$ |
> | ('Naphthalene', 'FPS')          | $92.8$          |
> | ('Salicylic Acid', 'Force')     | $6.5$           |
> | ('Salicylic Acid', 'Stability') | $281_{(37)}$    |
> | ('Salicylic Acid', 'h(r)')       | $0.03_{(0.00)}$ |
> | ('Salicylic Acid', 'FPS')       | $90.5$          |
>
> PaiNN on water-10k:
>
> |                       | PaiNN           |
> |:----------------------|:----------------|
> | Force                 | $5.1$           |
> | Stability             | $12_{(13)}$     |
> | ${RDF}_{(O,O)}$ | $0.30_{(0.14)}$ |
> | ${RDF}_{(H,H)}$ | $0.21_{(0.09)}$ |
> | ${RDF}_{(H,O)}$ | $0.29_{(0.12)}$ |
> | Diffusivity           | -           |
> | FPS                   | $71.8$          |
>
> ML for MD simulation is a fast-growing field, and we cannot possibly include every model. As our benchmark and codebase will be open-sourced (available now in supplementary), we encourage community contributions of more baseline models.

---

> > ### Author Response · Authors · 2022-11-11
> > **Response part 2**
> >
> > > More data are encouraged to include in this benchmark. For example, include all eight molecules from MD17. In addition, chignolin used in [2] is a peptide that is extensively studied with MD. It can be added to your peptide track. Moreover, polymers used in [3] could also be added to your material track.
> >
> > We appreciate the suggestion of adding more benchmark datasets. We note that training and simulating MD with ML force fields are computationally-intensive. Simulations reported in this paper take up to 75 hours for a single trajectory, as millions of steps are required to obtain meaningful results. The total GPU hours used in the paper’s simulations (not including training the models) exceeds 3,000 GPU hours. In addition, training the models uses 1000+ GPU hours. The main reason that we choose small systems for benchmarking is the high computational cost for MD simulation, as we wish to establish computationally accessible benchmarks to diverse users. Despite small sizes, the systems included in this paper are carefully chosen to be representative of a variety of real-world MD applications.
> >
> > With limited compute resources, we are only able to finish the experiments for PaiNN on MD17 and water, and a large water system (512 molecules) at this point. We are currently running SphereNet experiments. We will be happy to add more MD17 molecules to the benchmark when our compute resource becomes available.
> >
> > Chignolin is widely studied in the scientific community but is overly costly for atomic simulation using current ML FF methods. The Chignolin dataset provided in [2] is for coarse-grained simulation, which keeps only the alpha-carbon positions. There are many unique challenges in coarse-grained simulations. Our benchmark focuses on atomic simulations for the best clarity. Atomic simulation of Chignolin can be very expensive. The Chignolin molecule has 175 atoms without considering the solvent. To obtain meaningful results, nanosecond-level simulation is required, which can take more than a month to simulate for some ML methods. In this work, we aim to provide an accessible atomic simulation benchmark with reasonable compute requirements to the community. Systems like Chignolin would be a good system to study for coarse-graining and in future work.
> >
> > Polymers used in [5] are coarse-grained polymers with no explicit notion of atoms. Each polymer, on average, contains 889.5 beads, and the simulation requires 500 million steps. Significant coarse-graining and time-integration are used in [5] for scalable simulation of such large-scale systems. Simulations of this scale would take several years on GPUs to simulate for existing ML FFs and is therefore not suitable for the present benchmark.
> >
> > **In addition, we also added a new benchmark dataset with 512 water molecules in the updated draft** (more details can be found in our response to reviewer UiVb).

---

> > > ### Author Response · Authors · 2022-11-11
> > > **Response part 3**
> > >
> > > > Since this is a benchmark work without proposing any novel technical method, I would suggest submitting the paper to a benchmark track rather than ICLR main track.
> > >
> > > As ICLR does **NOT** have a benchmark track, many novel benchmark papers [6,7,8,9,10] have been accepted to the ICLR conference in past years. Benchmarks have been a significant component in the development of machine learning research, especially in bridging the gap between ML and the respective fields of application.
> > >
> > > We argue our benchmark, experimental analysis, and insights are **novel and significant contributions to the ML community.** Machine learning for MD simulation is a fast-growing field, but the force error-based benchmarks cannot reflect the practical utilization. We carefully curate representative systems of practical significance and reasonable compute requirements, and establish metrics with scientific grounding. We conduct thorough experiments to reveal the challenges and novel insights into ML for MD simulations, and we believe our benchmark is an important step in pushing the field forward. We have modified the paper to reflect the novelty aspects.
> > >
> > > We look forward to further discussion if you have additional feedback.
> > >
> > > Reference:
> > >
> > > [1] Liu, Yi, et al. "Spherical message passing for 3d molecular graphs." International Conference on Learning Representations. 2022.
> > >
> > > [2] Doerr, Stefan, et al. "Torchmd: A deep learning framework for molecular simulations." Journal of chemical theory and computation 17.4 (2021): 2355-2363.
> > >
> > > [3] Larsen, Ask Hjorth, et al. "The atomic simulation environment—a Python library for working with atoms." Journal of Physics: Condensed Matter 29.27 (2017): 273002.
> > >
> > > [4] Schütt, Kristof, Oliver Unke, and Michael Gastegger. "Equivariant message passing for the prediction of tensorial properties and molecular spectra." International Conference on Machine Learning. PMLR, 2021.
> > > [5] Fu, Xiang, et al. "Simulate Time-integrated Coarse-grained Molecular Dynamics with Geometric Machine Learning." arXiv preprint arXiv:2204.10348 (2022).
> > >
> > > [6] Santurkar, Shibani, Dimitris Tsipras, and Aleksander Madry. "Breeds: Benchmarks for subpopulation shift." International Conference on Learning Representations. 2021.
> > >
> > > [7] Li, Wenda, et al. "Isarstep: a benchmark for high-level mathematical reasoning." International Conference on Learning Representations. 2021.
> > >
> > > [8] Mehrjou, Arash, et al. "GeneDisco: A Benchmark for Experimental Design in Drug Discovery." International Conference on Learning Representations. 2022.
> > >
> > > [9] Wei, Jiaheng, et al. "Learning with noisy labels revisited: A study using real-world human annotations." International Conference on Learning Representations. 2022.
> > >
> > > [10] Sagawa, Shiori, et al. "Extending the wilds benchmark for unsupervised adaptation." International Conference on Learning Representations. 2022.

---

> > > > ### Author Response · Authors · 2022-11-16
> > > > **Response part 4: more experimental results**
> > > >
> > > > We have finished training and simulations of SphereNet on MD17. **The complete MD17 results are updated in  Table 3, Figure 3, and Figure 11.** We also summarize the results in the table below:
> > > >
> > > >
> > > > |                                  | DeepPot-SE      | SchNet          | DimeNet         | PaiNN           | SphereNet       | ForceNet        | GemNet-T        | GemNet-dT       | NequIP          |
> > > > |:---------------------------------|:----------------|:----------------|:----------------|:----------------|:----------------|:----------------|:----------------|:----------------|:----------------|
> > > > | ('Aspirin', 'Force')            | $21.0$          | $35.6$          | $10.0$          | $9.2$           | $3.4$           | $22.1$          | $3.3$           | $5.1$           | $2.3$           |
> > > > | ('Aspirin', 'Stability')        | $9_{(15)}$      | $26_{(23)}$     | $54_{(12)}$     | $159_{(121)}$   | $141_{(54)}$    | $182_{(144)}$   | $72_{(50)}$     | $192_{(132)}$   | $300_{(0)}$     |
> > > > | ('Aspirin', 'h(r)')              | $0.65_{(0.47)}$ | $0.36_{(0.57)}$ | $0.04_{(0.00)}$ | $0.04_{(0.01)}$ | $0.03_{(0.00)}$ | $0.56_{(0.15)}$ | $0.04_{(0.02)}$ | $0.04_{(0.01)}$ | $0.02_{(0.00)}$ |
> > > > | ('Aspirin', 'FPS')              | $88.0$          | $108.9$         | $20.6$          | $85.8$          | $17.5$          | $137.3$         | $28.2$          | $56.8$          | $8.4$           |
> > > > | ('Ethanol', 'Force')            | $8.9$           | $16.8$          | $4.2$           | $5.0$           | $1.7$           | $14.9$          | $2.1$           | $1.7$           | $1.3$           |
> > > > | ('Ethanol', 'Stability')        | $300_{(0)}$     | $247_{(106)}$   | $26_{(10)}$     | $86_{(109)}$    | $33_{(16)}$     | $300_{(0)}$     | $169_{(98)}$    | $300_{(0)}$     | $300_{(0)}$     |
> > > > | ('Ethanol', 'h(r)')              | $0.09_{(0.00)}$ | $0.21_{(0.11)}$ | $0.15_{(0.03)}$ | $0.15_{(0.08)}$ | $0.13_{(0.03)}$ | $0.86_{(0.05)}$ | $0.10_{(0.02)}$ | $0.09_{(0.00)}$ | $0.08_{(0.00)}$ |
> > > > | ('Ethanol', 'FPS')              | $101.0$         | $112.6$         | $21.4$          | $87.3$          | $30.5$          | $141.1$         | $27.1$          | $54.3$          | $8.9$           |
> > > > | ('Naphthalene', 'Force')        | $13.4$          | $22.5$          | $5.7$           | $3.8$           | $1.5$           | $9.9$           | $1.5$           | $1.9$           | $1.1$           |
> > > > | ('Naphthalene', 'Stability')    | $246_{(109)}$   | $18_{(2)}$      | $85_{(68)}$     | $300_{(0)}$     | $6_{(3)}$       | $300_{(0)}$     | $8_{(2)}$       | $25_{(10)}$     | $300_{(0)}$     |
> > > > | ('Naphthalene', 'h(r)')          | $0.11_{(0.00)}$ | $0.09_{(0.00)}$ | $0.10_{(0.01)}$ | $0.13_{(0.00)}$ | $0.14_{(0.04)}$ | $1.02_{(0.00)}$ | $0.13_{(0.00)}$ | $0.12_{(0.01)}$ | $0.12_{(0.01)}$ |
> > > > | ('Naphthalene', 'FPS')          | $109.3$         | $110.9$         | $19.1$          | $92.8$          | $18.3$          | $140.2$         | $27.7$          | $53.5$          | $8.2$           |
> > > > | ('Salicylic Acid', 'Force')     | $14.9$          | $26.3$          | $9.6$           | $6.5$           | $2.6$           | $12.8$          | $4.0$           | $4.0$           | $1.6$           |
> > > > | ('Salicylic Acid', 'Stability') | $300_{(0)}$     | $300_{(0)}$     | $73_{(82)}$     | $281_{(37)}$    | $36_{(16)}$     | $1_{(0)}$       | $26_{(24)}$     | $94_{(109)}$    | $300_{(0)}$     |
> > > > | ('Salicylic Acid', 'h(r)')       | $0.03_{(0.00)}$ | $0.03_{(0.00)}$ | $0.06_{(0.02)}$ | $0.03_{(0.00)}$ | $0.06_{(0.02)}$ | $0.35_{(0.00)}$ | $0.08_{(0.04)}$ | $0.07_{(0.03)}$ | $0.03_{(0.00)}$ |
> > > > | ('Salicylic Acid', 'FPS')       | $94.6$          | $111.7$         | $19.4$          | $90.5$          | $21.4$          | $143.2$         | $28.5$          | $52.4$          | $8.4$           |
> > > >
> > > >
> > > > SphereNet utilizes multi-hop information as it uses angle/torsion information in its spherical message passing. With this more complete representation of 3D molecular graphs, we observe that SphereNet achieves very good force prediction results. On the other hand, this flexible representation may also make SphereNet sensitive to out-of-distribution configurations — a potential cause of instability.
> > > >
> > > > We are currently running experiments for SphereNet on the water datasets and will be running SphereNet and PaiNN on the alanine dipeptide and the LiPS systems next. We will update more results as they become available.

---

> > > > > ### Comment · Reviewer_j9ZS · 2022-11-17
> > > > > **Reviewer Response**
> > > > >
> > > > > I thank the authors for the explanation and the additional experimental results. They have addressed most of my concerns. I would increase the score and have the following suggestions.
> > > > >
> > > > > 1. It would be beneficial to mention in the main text that this work only focuses on atomic-level simulation.
> > > > >
> > > > > 2. Describe the criterion of dataset selection and explain why other systems are not selected in the main text. Since coarse-grained simulation is quite common for larger systems, many people might be interested in such molecular systems.

---

> > > > > > ### Author Response · Authors · 2022-11-17
> > > > > > **Thank you for the reply and suggestions; updated draft**
> > > > > >
> > > > > > Thank you for the reply and the new suggestions.
> > > > > >
> > > > > > We have updated section 4, Dataset and Metrics, and section 7, Conclusion and Outlook, to state the focus of this paper on atomic-level simulations and the criterion of dataset selection. We also mention learning coarse-grained simulation as a future direction for accelerating the simulation of large systems such as polymers and proteins.
> > > > > >
> > > > > > May we know if there are still any specific issues that prevent you from recommending acceptance? We are more than happy to address them and further improve the paper!

---

> > > > > > > ### Comment · Reviewer_j9ZS · 2022-11-18
> > > > > > > **Reviewer Response**
> > > > > > >
> > > > > > > Thank you for further improvement! I will increase the score to 6.

---

> > > > > > > > ### Author Response · Authors · 2022-11-18
> > > > > > > > **Thank you**
> > > > > > > >
> > > > > > > > Thank you for helping improve our submission!

---

> ### Author Response · Authors · 2022-11-18
> **Response part 5: more experimental results -- SphereNet on Water**
>
> We have finished training and simulation of SphereNet on Water-10k and Water-large. **The results are updated in Figure 1, Table 4, Table 11, and Figure 13.** We also summarize the results below:
>
> Water-10k:
>
> |                       | DeepPot-SE      | SchNet          | DimeNet         | PaiNN           | SphereNet       | ForceNet        | GemNet-T        | GemNet-dT       | NequIP          |
> |:----------------------|:----------------|:----------------|:----------------|:----------------|:----------------|:----------------|:----------------|:----------------|:----------------|
> | Force                 | $5.8$           | $9.5$           | $1.4$           | $5.1$           | $16.1$          | $10.9$          | $0.7$           | $1.3$           | $1.5$           |
> | Stability             | $247_{(147)}$   | $232_{(59)}$    | $30_{(10)}$     | $12_{(13)}$     | $500_{(0)}$     | $7_{(3)}$       | $25_{(7)}$      | $7_{(3)}$       | $500_{(0)}$     |
> | ${RDF}_{(O,O)}$ | $0.07_{(0.01)}$ | $0.63_{(0.04)}$ | $0.27_{(0.15)}$ | $0.30_{(0.14)}$ | $0.89_{(0.04)}$ | $0.79_{(0.03)}$ | $0.22_{(0.05)}$ | $0.42_{(0.22)}$ | $0.06_{(0.02)}$ |
> | ${RDF}_{(H,H)}$ | $0.06_{(0.02)}$ | $0.30_{(0.02)}$ | $0.18_{(0.08)}$ | $0.21_{(0.09)}$ | $0.40_{(0.01)}$ | $0.55_{(0.01)}$ | $0.16_{(0.03)}$ | $0.35_{(0.25)}$ | $0.05_{(0.01)}$ |
> | ${RDF}_{(H,O)}$ | $0.19_{(0.05)}$ | $0.57_{(0.04)}$ | $0.21_{(0.04)}$ | $0.29_{(0.12)}$ | $1.14_{(0.03)}$ | $1.34_{(0.03)}$ | $0.20_{(0.04)}$ | $0.42_{(0.27)}$ | $0.27_{(0.07)}$ |
> | Diffusivity           | $0.04$          | $1.90$          | -           | -           | $2.23$          | -           | -           | -           | $0.18$          |
> | FPS                   | $91.0$          | $78.9$          | $17.9$          | $71.8$          | $3.1$           | $67.6$          | $11.3$          | $33.7$          | $3.9$           |
>
> Water-large (*SphereNet on Water-large requires more memory than Tesla V100. We run its simulation on faster NVIDIA A100 cards, so the FPS is not entirely comparable to other models.):
> |                       | DeepPot-SE      | SchNet          | DimeNet         | PaiNN           | SphereNet*       | ForceNet        | GemNet-T        | GemNet-dT       | NequIP          |
> |:----------------------|:----------------|:----------------|:----------------|:----------------|:----------------|:----------------|:----------------|:----------------|:----------------|
> | Force                 | $10.6$          | $12.1$          | $5.1$           | $9.7$           | $18.4$          | $13.2$          | $5.6$           | $4.2$           | $7.7$           |
> | Stability             | $19_{(22)}$     | $118_{(58)}$    | $38_{(13)}$     | $16_{(12)}$     | $150_{(0)}$     | $8_{(0)}$       | $45_{(25)}$     | $50_{(9)}$      | $150_{(0)}$     |
> | ${RDF}_{(O,O)}$ | $0.23_{(0.06)}$ | $0.62_{(0.01)}$ | $0.17_{(0.03)}$ | $0.31_{(0.06)}$ | $0.93_{(0.02)}$ | $0.74_{(0.02)}$ | $0.22_{(0.16)}$ | $0.16_{(0.02)}$ | $0.10_{(0.01)}$ |
> | ${RDF}_{(H,H)}$ | $0.24_{(0.06)}$ | $0.30_{(0.04)}$ | $0.12_{(0.03)}$ | $0.21_{(0.05)}$ | $0.42_{(0.01)}$ | $0.51_{(0.02)}$ | $0.15_{(0.11)}$ | $0.11_{(0.01)}$ | $0.07_{(0.00)}$ |
> | ${RDF}_{(H,O)}$ | $0.67_{(0.27)}$ | $0.55_{(0.01)}$ | $0.17_{(0.02)}$ | $0.29_{(0.05)}$ | $0.97_{(0.03)}$ | $1.38_{(0.05)}$ | $0.23_{(0.12)}$ | $0.16_{(0.02)}$ | $0.12_{(0.02)}$ |
> | Diffusivity           | -           | $2.54$          | -           | -           | $2.98$          | -           | -           | -           | $0.89$          |
> | FPS                   | $80.7$          | $23.1$          | $3.5$           | $17.4$          | $0.8$           | $11.9$          | $2.2$           | $5.3$           | $0.7$           |
>
> As SphereNet uses multi-hop information (angle/torsion) in its message passing, it suffers from scalability and has lower FPS in a condensed-phase system like our water system. This also influences training – given our 1-week training time budget on a Tesla V100 GPU, SphereNet has fewer training epochs compared to more efficient models. Interestingly, SphereNet does not perform well on the water datasets but is very stable. However, it does not produce accurate ensemble properties. Its RDF curves (in Figure 13) show that the dynamics it learns are stable but incorrect. We hypothesize that the angular and torsional information used in SphereNet are great inductive biases for organic molecules, where such interactions are important. It may not be suitable for condensed phase systems with very simple molecules like water.
>
> We are currently running PaiNN and SphereNet experiments on other water dataset sizes, alanine dipeptide, and LiPS. We will update the results once they become available.

---

> ### Author Response · Authors · 2022-11-25
> **More experimental results (1)**
>
> We have finished running experiments of PaiNN and SphereNet on LiPS, Alanine dipeptide, water-1k, and water-90k. We summarize the results below. We can not update the paper as of now and will update the results (Figure 4, Table 5, Table 6, Table 8, Table 9, Figure 9, Figure 12, Figure 14, Figure 15)in the final version.
>
> Overall, both PaiNN and SphereNet are not stable enough for the alanine dipeptide task. On the LiPS dataset, both PaiNN and SphereNet are not the best performing model in force error but can simulate stably with reasonable diffusivity estimation.
>
>
> Alanine dipeptide:
>
> |                     | DeepPot-SE                 | SchNet                       | DimeNet                      | PaiNN                        | SphereNet                    | ForceNet                     | GemNet-T                     | GemNet-dT                    | NequIP                         |
> |:--------------------|:---------------------------|:-----------------------------|:-----------------------------|:-----------------------------|:-----------------------------|:-----------------------------|:-----------------------------|:-----------------------------|:-------------------------------|
> | Force MAE           | $272.1$      | $217.0$       | $239.0$     | $266.2$        | $256.3$  | $284.7$        | $233.5$     | $219.7$       | $215.6$         |
> | #Finished           | 0/6          | 0/6            | 0/6            | 0/6            | 0/6            | 0/6            | 0/6            | 0/6            | 5/6        |
> | Stability           | $0_{(0)}$    | $0_{(0)}$      | $0_{(0)}$      | $0_{(0)}$      | $4_{(7)}$      | $0_{(0)}$      | $18_{(27)}$    | $0_{(0)}$      | $4168_{(1860)}$  |
> | ${PMF}_{\phi}$ | -  | -  | -  | -  | -  | -  | -  | -  | $108_{(2)}$       |
> | ${PMF}_{\psi}$ | -  | -  | -  | -  | -  | -  | -  | -  | $126_{(4)}$       |
> | FPS                 | $54.3$      | $42.4$      | $12.1$         | $42.2$      | $9.9$          | $99.1$        | $15.0$         | $36.5$      | $8.3$            |
>
>
> LiPS:
>
> |             | DeepPot-SE                     | SchNet                          | DimeNet                         | PaiNN                           | SphereNet                       | ForceNet                       | GemNet-T                        | GemNet-dT                       | NequIP                          |
> |:------------|:-------------------------------|:--------------------------------|:--------------------------------|:--------------------------------|:--------------------------------|:-------------------------------|:--------------------------------|:--------------------------------|:--------------------------------|
> | Force       | $40.5$           | $28.8$            | $3.2$          | $11.7$        | $8.3$         | $12.8$       | $1.3$            | $1.4$            | $3.7$          |
> | Stability   | $4_{(3)}$        | $50_{(0)}$       | $48_{(4)}$       | $50_{(0)}$       | $50_{(0)}$       | $26_{(8)}$    | $50_{(0)}$       | $50_{(0)}$       | $50_{(0)}$       |
> | RDF         | $0.27_{(0.15)}$  | $0.04_{(0.00)}$  | $0.05_{(0.01)}$  | $0.04_{(0.01)}$  | $0.04_{(0.00)}$  | $0.51_{(0.08)}$  | $0.04_{(0.00)}$  | $0.04_{(0.00)}$  | $0.04_{(0.01)}$  |
> | Diffusivity | -            | $0.38$        | $0.30$         | $0.40$        | $0.40$        | -            | $0.24$           | $0.28$           | $0.34$         |
> | FPS         | $66.1$          | $35.2$         | $14.8$        | $75.7$           | $18.1$        | $72.1$          | $16.9$        | $43.5$           | $8.2$             |

---

> > ### Author Response · Authors · 2022-11-25
> > **More experimental results (2)**
> >
> > For the water datasets, we see PaiNN is not stable. SphereNet is stable but produces incorrect ensemble statistics. Similar to what we have observed in Water-10k, SphereNet's multi-hop message passing is inefficient for condensed-phased systems, and FPS is low.
> >
> > Water-1k:
> >
> > |                       | DeepPot-SE      | SchNet          | DimeNet         | PaiNN           | SphereNet       | ForceNet        | GemNet-T        | GemNet-dT     | NequIP          |
> > |:----------------------|:----------------|:----------------|:----------------|:----------------|:----------------|:----------------|:----------------|:--------------|:----------------|
> > | Force                 | $6.7$           | $13.1$          | $3.5$           | $5.2$           | $27.5$          | $13.6$          | $5.0$           | $29.2$        | $1.4$           |
> > | Stability             | $108_{(117)}$   | $175_{(56)}$    | $4_{(4)}$       | $14_{(9)}$      | $385_{(160)}$   | $13_{(7)}$      | $6_{(7)}$       | $0_{(0)}$     | $500_{(0)}$     |
> > | ${RDF}_{(O,O)}$ | $0.17_{(0.10)}$ | $0.52_{(0.05)}$ | $0.46_{(0.22)}$ | $0.21_{(0.05)}$ | $0.65_{(0.02)}$ | $0.86_{(0.09)}$ | $0.62_{(0.48)}$ | - | $0.07_{(0.02)}$ |
> > | ${RDF}_{(H,H)}$ | $0.13_{(0.09)}$ | $0.24_{(0.02)}$ | $0.33_{(0.15)}$ | $0.15_{(0.04)}$ | $0.29_{(0.01)}$ | $0.56_{(0.04)}$ | $0.35_{(0.21)}$ | - | $0.07_{(0.02)}$ |
> > | ${RDF}_{(H,O)}$ | $0.28_{(0.15)}$ | $0.54_{(0.01)}$ | $0.43_{(0.17)}$ | $0.16_{(0.04)}$ | $0.81_{(0.03)}$ | $1.44_{(0.09)}$ | $0.71_{(0.65)}$ | - | $0.26_{(0.07)}$ |
> > | Diffusivity           | $0.24$          | $1.79$          | -           | -           | $2.05$          | -           | -           | -         | $0.37$          |
> > | FPS                   | $61.8$          | $99.2$          | $16.4$          | $54.5$          | $3.0$           | $68.1$          | $15.4$          | $34.5$        | $3.9$           |
> >
> > Water-10k:
> >
> > |                       | DeepPot-SE      | SchNet          | DimeNet         | PaiNN           | SphereNet       | ForceNet        | GemNet-T        | GemNet-dT       | NequIP          |
> > |:----------------------|:----------------|:----------------|:----------------|:----------------|:----------------|:----------------|:----------------|:----------------|:----------------|
> > | Force                 | $5.9$           | $8.4$           | $1.7$           | $5.1$           | $14.8$          | $8.6$           | $0.7$           | $1.1$           | $1.4$           |
> > | Stability             | $500_{(0)}$     | $299_{(70)}$    | $36_{(9)}$      | $16_{(7)}$      | $500_{(0)}$     | $9_{(12)}$      | $20_{(9)}$      | $8_{(10)}$      | $500_{(0)}$     |
> > | ${RDF}_{(O,O)}$ | $0.07_{(0.02)}$ | $0.67_{(0.03)}$ | $0.21_{(0.03)}$ | $0.26_{(0.17)}$ | $0.91_{(0.04)}$ | $1.31_{(0.49)}$ | $0.35_{(0.23)}$ | $0.20_{(0.01)}$ | $0.06_{(0.01)}$ |
> > | ${RDF}_{(H,H)}$ | $0.05_{(0.01)}$ | $0.31_{(0.02)}$ | $0.14_{(0.01)}$ | $0.20_{(0.12)}$ | $0.42_{(0.03)}$ | $0.82_{(0.26)}$ | $0.25_{(0.19)}$ | $0.16_{(0.01)}$ | $0.04_{(0.01)}$ |
> > | ${RDF}_{(H,O)}$ | $0.29_{(0.08)}$ | $0.67_{(0.04)}$ | $0.18_{(0.02)}$ | $0.21_{(0.06)}$ | $1.24_{(0.08)}$ | $2.05_{(0.60)}$ | $0.24_{(0.06)}$ | $0.26_{(0.02)}$ | $0.25_{(0.06)}$ |
> > | Diffusivity           | $0.35$          | $1.97$          | -           | -           | $2.26$          | -           | -           | -           | $0.18$          |
> > | FPS                   | $62.1$          | $103.0$         | $16.3$          | $71.9$          | $3.1$           | $43.8$          | $15.3$          | $32.7$          | $3.0$           |

---

### Official Review · Reviewer_NbtG · 2022-10-24

**Confidence:** 4
**Correctness:** 3
**Technical Novelty And Significance:** 2
**Empirical Novelty And Significance:** 3
**Recommendation:** 5

**Clarity, Quality, Novelty And Reproducibility:**

The paper is well written, and the clarity is good.

About originality, the main original parts are the new evaluation metrics and the experimental analysis. But the used datasets and methods are mainly proposed by previous work.


**Strength And Weaknesses:**

> Strengths

1. This paper is well-written and easy to follow. The motivation, contribution, etc. are clear.
2. The authors point out that forces are not enough to evaluate MD and propose several new metrics. Based on my understanding, the evaluation metrics are new, and the inconsistent performance on different metrics may inspire more future work.

> Weaknesses

1. About **the datasets** in this benchmark:

     - Based on my understanding, OC20 can also be included in this benchmark since it also includes complete relaxation trajectories  (https://github.com/Open-Catalyst-Project/ocp/blob/main/DATASET.md) and is widely used recently. The authors should justify why they don't include this dataset. In addition, if considering OC20, which metric should we use to evaluate the model?
    - The size of the systems is limited to less than 200 atoms. This may be not suitable for many real-world datasets like water, goop, and sand in [1] and protein data.


2. I have several questions about **the baseline methods** in Table 2.

    - I think there is an issue with GemNet-T and GemNet-dT. GemNet-T should be energy-conserving, while GemNet-dT is not.
    - Based on my knowledge, except NequIP, all the other baseline methods in Table 2 are invariant to rotation, and/or translation, and/or reflection. Different from GemNet-T, GemNet-dT uses a different way to output forces.
    - Since the task is to predict forces (should be equivariant), I suggest the authors include more equivariant baselines like [2] and [3].

3. For **the detailed evaluation metrics** in Appendix A, it is unclear to me which is used in previous work and which is newly proposed in this paper. This is because some metrics have corresponding references, and some do not.

[1] Learning to Simulate Complex Physics with Graph Networks

[2] Equivariant message passing for the prediction of tensorial properties and molecular spectra

[3] E(n) Equivariant Graph Neural Networks


**Summary Of The Paper:**

This work focuses on molecular dynamics (MD) simulations. Previous methods use force/energy errors as the evaluation, but based on the authors' analysis, these metrics are not enough.

Specifically, in this paper, the authors first curate four MD datasets based on previous work and design specific evaluation metrics for each dataset. They then run experiments with seven baseline methods. Finally, they do a lot of analysis on the results, showing forces are not enough, and providing potential solutions.


**Summary Of The Review:**

This paper is well-written and easy to follow. The motivation, contribution, etc are clear. The experimental analysis looks good.

But there exist several weaknesses.

- My main concerns are the used **datasets and baseline methods**. OC20 dataset and more equivariant baseline methods should be considered.

- There are some errors in Table 2 (**baseline methods**).

- For **the evaluation metrics**, it is unclear to me which is used in previous work and which is newly proposed in this paper.

In addition, the main contributions are the benchmark, experimental analysis, and some insights. I think it might be more suitable for a benchmark track.

---

> ### Author Response · Authors · 2022-11-11
> **Response part 1**
>
> We thank reviewer NbtG for helpful questions and suggestions. We address each of the reviewer’s concerns below.
>
> > The authors should justify why they don't include this dataset. In addition, if considering OC20, which metric should we use to evaluate the model?
>
> Thank you for the question. We briefly mentioned the difference between our benchmark and the OC benchmark in section 2, and we agree it is important to clarify further the difference between our benchmark and other existing benchmarks. In short, OC20/22 are NOT molecular dynamics datasets but structural relaxation datasets.
>
> The goal of OC20\/22 is to relax structures to their equilibrium state and predict the final energy, but they don’t evaluate any dynamical properties. In contrast, our benchmark focuses on the stability of running MD and accurately predicting the ensemble properties of atomic systems.
>
> The existing OC20 and OC22 datasets only include structure relaxation (hundreds of steps), not MD trajectories that sample many different configurations (millions of steps). Extending the OC20/22 dataset to study MD requires 2-4 orders of magnitude more compute than OC20/22, which is not feasible for us.
>
> The metrics originally proposed in OC are suitable and are well-benchmarked by the original paper. The dataset is not suitable for a benchmark on molecular dynamics simulation. We have modified the related works section to reflect these distinctions.
>
> > The size of the systems is limited to less than 200 atoms.
>
> We agree with the reviewer that it is valuable to study larger systems. In the updated draft, we added a new benchmark dataset with 512 water molecules (1536 atoms) simulated with the same reference force field (SPC/E-fw).
>
> We evaluate ML FF models trained on water-10k to simulate the 512 water molecules for 150 ps, and compute the same metrics as the small water (64 molecules) dataset: force error, stability, RDF, and diffusivity. **We add a paragraph of discussion and Table 11 to Appendix C where we demonstrate the experiment results.** We also summarize this result in the table below. One standard deviation for applicable metrics is in subscript. The computation of diffusivity requires 100 ps stable simulation.
>
> We observe that all models suffer slightly higher force errors compared to evaluation over the 64-molecule water system. In terms of stability, NequIP always remains stable for the entire 150 ps. SchNet is the second most stable model, while all other models are not stable enough for diffusivity computation. DimeNet, GemNet, and GemNet-dT are not stable through the entire simulation, but can produce decent RDF results.
>
> |                       | DeepPot-SE      | SchNet          | DimeNet         | PaiNN           | ForceNet        | GemNet-T        | GemNet-dT       | NequIP          |
> |:----------------------|:----------------|:----------------|:----------------|:----------------|:----------------|:----------------|:----------------|:----------------|
> | Force                 | $10.6$          | $12.1$          | $5.1$           | $9.7$           | $13.2$          | $5.6$           | $4.2$           | $7.7$           |
> | Stability             | $19_{(22)}$     | $118_{(58)}$    | $38_{(13)}$     | $16_{(12)}$     | $8_{(0)}$       | $45_{(25)}$     | $50_{(9)}$      | $150_{(0)}$     |
> | ${RDF}_{(O,O)}$ | $0.23_{(0.06)}$ | $0.62_{(0.01)}$ | $0.17_{(0.03)}$ | $0.31_{(0.06)}$ | $0.74_{(0.02)}$ | $0.22_{(0.16)}$ | $0.16_{(0.02)}$ | $0.10_{(0.01)}$ |
> | ${RDF}_{(H,H)}$ | $0.24_{(0.06)}$ | $0.30_{(0.04)}$ | $0.12_{(0.03)}$ | $0.21_{(0.05)}$ | $0.51_{(0.02)}$ | $0.15_{(0.11)}$ | $0.11_{(0.01)}$ | $0.07_{(0.00)}$ |
> | ${RDF}_{(H,O)}$ | $0.67_{(0.27)}$ | $0.55_{(0.01)}$ | $0.17_{(0.02)}$ | $0.29_{(0.05)}$ | $1.38_{(0.05)}$ | $0.23_{(0.12)}$ | $0.16_{(0.02)}$ | $0.12_{(0.02)}$ |
> | Diffusivity           | -           | $2.54$          | -           | -           | -           | -           | -           | $0.89$          |
> | FPS                   | $80.7$          | $23.1$          | $3.5$           | $17.4$          | $11.9$          | $2.2$           | $5.3$           | $0.7$           |
>
> We note that the large water system is very expensive to simulate. It requires 62 GPU hours to simulate a single 150 ps trajectory using NequIP on a Tesla V100 GPU. The main reason that we choose small systems for benchmarking is the high computational cost for MD simulation, as we wish to establish computationally accessible benchmarks to diverse users. Simulations reported in this paper take up to 75 hours for a single trajectory, as millions of steps are required to obtain meaningful results. The total GPU hours used in the paper’s experiments (not including training the models) exceeds 3,000 GPU hours. In addition, training the models uses 1000+ GPU hours. Despite small sizes, the systems included in this paper are carefully chosen to be representative of a variety of real-world MD applications.

---

> > ### Author Response · Authors · 2022-11-11
> > **Response part 2**
> >
> > > This may be not suitable for many real-world datasets like water, goop, and sand in [1] and protein data.
> >
> > The water, goop, and sand datasets described in [1] are completely different data/tasks compared to MD simulations. Datasets in [1] are generated through material point method (MPM) calculations. The systems are at real-world length/time scales in the units of meter and second, and MPM uses a homogenized representation of the corresponding particle-based system. The simulated trajectories in [1] are within 1000 time steps. MD simulations are at microscopic length/time scales in the units of nanometer and nanosecond. Meaningful simulations often require millions of time steps. The particle interactions of datasets in [1] (uniform bead types, local interactions) are also much simpler than MD (many different atom types, complex bonded, non-bonded, angular, long-range, etc. interactions). The task is different, too: models in [1] predict acceleration, and ML FF models predict energy/forces.
> >
> > Protein is of high interest to the MD simulation community. However, the spatial/temporal complexity required for protein simulation is often prohibitive. Protein dynamics often require very long simulations at microsecond-level (billions of steps) to characterize and requires significant effort to simulate even with highly-efficient traditional methods and specialized hardware [2]. Simulating meaningful trajectories of proteins using existing ML FFs is unfeasible (would take years on GPUs) for a benchmark. This work aims to accelerate the development of ML methods for MD simulation by proposing systems and metrics that are representative of real-world applications and can be benchmarked with reasonable time and computation.
> >
> > > There are some errors in Table 2 (baseline methods).
> >
> > Thank you for pointing out an error in Table 2 and we have corrected it in the updated draft. GemNet-T is energy conserving and GemNet-dT is not. The difference is GemNet-T predicts forces by taking the derivative of energy with respect to atom positions (which ensures energy conservation), while GemNet-dT directly predicts the forces for each atom (and therefore does not conserve energy). There is a detailed discussion on this distinction in Section 5 and Figure 5(d). We believe other information in Table 2 is correct. As stated in the original papers[3,4]: ForceNet is not invariant to rotation/reflection, and GemNet is SE(3)-equivariant.
> >
> > > Since the task is to predict forces (should be equivariant), I suggest the authors include more equivariant baselines like [2] and [3].
> >
> > We appreciate the suggestion of adding more baseline models. We are running experiments for PaiNN [5] and SphereNet [6]. With limited compute resources, we are only able to finish the experiments for the large water system (512 molecules), and PaiNN on MD17 and water at this point. We are currently running SphereNet experiments and will add the results when they become available.
> >
> > **We have added PaiNN to Figure1, Figure 3, Table 2, Table 3, and Table 4.** We will update more results as they become available. We also summarize the results in the table below. PaiNN attains decent performance on MD17 molecules but is not stable for water. One standard deviation for applicable metrics is in subscript. The computation of diffusivity requires 100 ps stable simulation.
> >
> > PaiNN on MD17:
> >
> > |                                  | PaiNN           |
> > |:---------------------------------|:----------------|
> > | ('Aspirin', 'Force')            | $9.2$           |
> > | ('Aspirin', 'Stability')        | $159_{(121)}$   |
> > | ('Aspirin', 'h(r)')              | $0.04_{(0.01)}$ |
> > | ('Aspirin', 'FPS')              | $85.8$          |
> > | ('Ethanol', 'Force')            | $5.0$           |
> > | ('Ethanol', 'Stability')        | $86_{(109)}$    |
> > | ('Ethanol', 'h(r)')              | $0.15_{(0.08)}$ |
> > | ('Ethanol', 'FPS')              | $87.3$          |
> > | ('Naphthalene', 'Force')        | $3.8$           |
> > | ('Naphthalene', 'Stability')    | $300_{(0)}$     |
> > | ('Naphthalene', 'h(r)')          | $0.13_{(0.00)}$ |
> > | ('Naphthalene', 'FPS')          | $92.8$          |
> > | ('Salicylic Acid', 'Force')     | $6.5$           |
> > | ('Salicylic Acid', 'Stability') | $281_{(37)}$    |
> > | ('Salicylic Acid', 'h(r)')       | $0.03_{(0.00)}$ |
> > | ('Salicylic Acid', 'FPS')       | $90.5$          |
> >
> >
> > PaiNN on water-10k:
> >
> > |                       | PaiNN           |
> > |:----------------------|:----------------|
> > | Force                 | $5.1$           |
> > | Stability             | $12_{(13)}$     |
> > | ${RDF}_{(O,O)}$ | $0.30_{(0.14)}$ |
> > | ${RDF}_{(H,H)}$ | $0.21_{(0.09)}$ |
> > | ${RDF}_{(H,O)}$ | $0.29_{(0.12)}$ |
> > | Diffusivity           | -           |
> > | FPS                   | $71.8$          |

---

> > > ### Author Response · Authors · 2022-11-11
> > > **Response part 3**
> > >
> > > PaiNN was already referenced in the paper, and we have added EGNN [7] to the reference. ML for MD simulation is a fast-growing field, and we cannot possibly include every model. As our benchmark and codebase will be open-sourced (available now in supplementary), we encourage community contributions of more baseline models in the future.
> > >
> > > > For the evaluation metrics, it is unclear to me which is used in previous work and which is newly proposed in this paper.
> > >
> > > All our benchmark metrics are **well-established in the scientific community, but are novel to the ML community.** The benchmark observables are widely used by researchers in the computational physics/chemistry/biology communities, while we are the first to propose a benchmark for machine learning methods utilizing simulations and these observables, in contrast to previous benchmarks based on force error. The stability criterion used in this paper is newly proposed based on the established observables.
> > >
> > > We believe the benchmark metrics should be familiar to the scientific community instead of something completely new: the ultimate goal of developing ML force fields is to use them in realistic scientific research. It is thus crucial to demonstrate that ML methods can recover the important metrics of interest to the respective community, and our benchmark is a step towards bridging the gap between the ML and scientific communities. We also added references for the benchmark metrics to Appendix A.
> > >
> > > > In addition, the main contributions are the benchmark, experimental analysis, and some insights. I think it might be more suitable for a benchmark track.
> > >
> > > As ICLR does **NOT** have a benchmark track, many novel benchmark papers [8,9,10,11,12] have been accepted to the ICLR conference in past years. Benchmarks have been a significant component in the development of machine learning research, especially in bridging the gap between ML and the respective fields of application.
> > >
> > > We argue our benchmark, experimental analysis, and insights are **novel and significant contributions to the ML community.** Machine learning for MD simulation is a fast-growing field, but the force error-based benchmarks cannot reflect practical utilization. We carefully curate representative systems of practical significance and reasonable compute requirements and establish metrics with scientific grounding. We conduct thorough experiments to reveal the challenges and novel insights into ML for MD simulations, and we believe our benchmark is an important step in pushing the field forward. We have modified the paper to reflect the novelty aspects. For a comparison between our work and some recent related works, we refer reviewer NbtG to our response to reviewer CmqU.
> > >
> > > We look forward to further discussion if you have additional feedback.
> > >
> > > Reference:
> > >
> > > [1] Sanchez-Gonzalez, Alvaro, et al. "Learning to simulate complex physics with graph networks." International Conference on Machine Learning. PMLR, 2020.
> > >
> > > [2] Lindorff-Larsen, Kresten, et al. "How fast-folding proteins fold." Science 334.6055 (2011): 517-520.
> > >
> > > [3] Gasteiger, Johannes, Florian Becker, and Stephan Günnemann. "Gemnet: Universal directional graph neural networks for molecules." Advances in Neural Information Processing Systems 34 (2021): 6790-6802.
> > >
> > > [4] Hu, Weihua, et al. "Forcenet: A graph neural network for large-scale quantum calculations." arXiv preprint arXiv:2103.01436 (2021).
> > >
> > > [5] Schütt, Kristof, Oliver Unke, and Michael Gastegger. "Equivariant message passing for the prediction of tensorial properties and molecular spectra." International Conference on Machine Learning. PMLR, 2021.
> > >
> > > [6] Liu, Yi, et al. "Spherical message passing for 3d molecular graphs." International Conference on Learning Representations. 2022.
> > >
> > > [7] Satorras, Vıctor Garcia, Emiel Hoogeboom, and Max Welling. "E (n) equivariant graph neural networks." International conference on machine learning. PMLR, 2021.
> > >
> > > [8] Santurkar, Shibani, Dimitris Tsipras, and Aleksander Madry. "Breeds: Benchmarks for subpopulation shift." International Conference on Learning Representations. 2021.
> > >
> > > [9] Li, Wenda, et al. "Isarstep: a benchmark for high-level mathematical reasoning." International Conference on Learning Representations. 2021.
> > >
> > > [10] Mehrjou, Arash, et al. "GeneDisco: A Benchmark for Experimental Design in Drug Discovery." International Conference on Learning Representations. 2022.
> > >
> > > [11] Wei, Jiaheng, et al. "Learning with noisy labels revisited: A study using real-world human annotations." International Conference on Learning Representations. 2022.
> > >
> > > [12] Sagawa, Shiori, et al. "Extending the wilds benchmark for unsupervised adaptation." International Conference on Learning Representations. 2022.

---

> > > > ### Author Response · Authors · 2022-11-16
> > > > **Response part 4: more experimental results**
> > > >
> > > > We have finished training and simulations of SphereNet on MD17. **The complete MD17 results are updated in  Table 3, Figure 3, and Figure 11.** We also summarize the results in the table below:
> > > >
> > > >
> > > > |                                  | DeepPot-SE      | SchNet          | DimeNet         | PaiNN           | SphereNet       | ForceNet        | GemNet-T        | GemNet-dT       | NequIP          |
> > > > |:---------------------------------|:----------------|:----------------|:----------------|:----------------|:----------------|:----------------|:----------------|:----------------|:----------------|
> > > > | ('Aspirin', 'Force')            | $21.0$          | $35.6$          | $10.0$          | $9.2$           | $3.4$           | $22.1$          | $3.3$           | $5.1$           | $2.3$           |
> > > > | ('Aspirin', 'Stability')        | $9_{(15)}$      | $26_{(23)}$     | $54_{(12)}$     | $159_{(121)}$   | $141_{(54)}$    | $182_{(144)}$   | $72_{(50)}$     | $192_{(132)}$   | $300_{(0)}$     |
> > > > | ('Aspirin', 'h(r)')              | $0.65_{(0.47)}$ | $0.36_{(0.57)}$ | $0.04_{(0.00)}$ | $0.04_{(0.01)}$ | $0.03_{(0.00)}$ | $0.56_{(0.15)}$ | $0.04_{(0.02)}$ | $0.04_{(0.01)}$ | $0.02_{(0.00)}$ |
> > > > | ('Aspirin', 'FPS')              | $88.0$          | $108.9$         | $20.6$          | $85.8$          | $17.5$          | $137.3$         | $28.2$          | $56.8$          | $8.4$           |
> > > > | ('Ethanol', 'Force')            | $8.9$           | $16.8$          | $4.2$           | $5.0$           | $1.7$           | $14.9$          | $2.1$           | $1.7$           | $1.3$           |
> > > > | ('Ethanol', 'Stability')        | $300_{(0)}$     | $247_{(106)}$   | $26_{(10)}$     | $86_{(109)}$    | $33_{(16)}$     | $300_{(0)}$     | $169_{(98)}$    | $300_{(0)}$     | $300_{(0)}$     |
> > > > | ('Ethanol', 'h(r)')              | $0.09_{(0.00)}$ | $0.21_{(0.11)}$ | $0.15_{(0.03)}$ | $0.15_{(0.08)}$ | $0.13_{(0.03)}$ | $0.86_{(0.05)}$ | $0.10_{(0.02)}$ | $0.09_{(0.00)}$ | $0.08_{(0.00)}$ |
> > > > | ('Ethanol', 'FPS')              | $101.0$         | $112.6$         | $21.4$          | $87.3$          | $30.5$          | $141.1$         | $27.1$          | $54.3$          | $8.9$           |
> > > > | ('Naphthalene', 'Force')        | $13.4$          | $22.5$          | $5.7$           | $3.8$           | $1.5$           | $9.9$           | $1.5$           | $1.9$           | $1.1$           |
> > > > | ('Naphthalene', 'Stability')    | $246_{(109)}$   | $18_{(2)}$      | $85_{(68)}$     | $300_{(0)}$     | $6_{(3)}$       | $300_{(0)}$     | $8_{(2)}$       | $25_{(10)}$     | $300_{(0)}$     |
> > > > | ('Naphthalene', 'h(r)')          | $0.11_{(0.00)}$ | $0.09_{(0.00)}$ | $0.10_{(0.01)}$ | $0.13_{(0.00)}$ | $0.14_{(0.04)}$ | $1.02_{(0.00)}$ | $0.13_{(0.00)}$ | $0.12_{(0.01)}$ | $0.12_{(0.01)}$ |
> > > > | ('Naphthalene', 'FPS')          | $109.3$         | $110.9$         | $19.1$          | $92.8$          | $18.3$          | $140.2$         | $27.7$          | $53.5$          | $8.2$           |
> > > > | ('Salicylic Acid', 'Force')     | $14.9$          | $26.3$          | $9.6$           | $6.5$           | $2.6$           | $12.8$          | $4.0$           | $4.0$           | $1.6$           |
> > > > | ('Salicylic Acid', 'Stability') | $300_{(0)}$     | $300_{(0)}$     | $73_{(82)}$     | $281_{(37)}$    | $36_{(16)}$     | $1_{(0)}$       | $26_{(24)}$     | $94_{(109)}$    | $300_{(0)}$     |
> > > > | ('Salicylic Acid', 'h(r)')       | $0.03_{(0.00)}$ | $0.03_{(0.00)}$ | $0.06_{(0.02)}$ | $0.03_{(0.00)}$ | $0.06_{(0.02)}$ | $0.35_{(0.00)}$ | $0.08_{(0.04)}$ | $0.07_{(0.03)}$ | $0.03_{(0.00)}$ |
> > > > | ('Salicylic Acid', 'FPS')       | $94.6$          | $111.7$         | $19.4$          | $90.5$          | $21.4$          | $143.2$         | $28.5$          | $52.4$          | $8.4$           |
> > > >
> > > >
> > > > SphereNet utilizes multi-hop information as it uses angle/torsion information in its spherical message passing. With this more complete representation of 3D molecular graphs, we observe that SphereNet achieves very good force prediction results. On the other hand, this flexible representation may also make SphereNet sensitive to out-of-distribution configurations — a potential cause of instability.
> > > >
> > > > We are currently running experiments for SphereNet on the water datasets and will be running SphereNet and PaiNN on the alanine dipeptide and the LiPS systems next. We will update more results as they become available.

---

> > > > > ### Author Response · Authors · 2022-11-18
> > > > > **Response part 5: more experimental results -- SphereNet on Water**
> > > > >
> > > > > We have finished training and simulation of SphereNet on Water-10k and Water-large. **The results are updated in Figure 1, Table 4, Table 11, and Figure 13.** We also summarize the results below:
> > > > >
> > > > > Water-10k:
> > > > >
> > > > > |                       | DeepPot-SE      | SchNet          | DimeNet         | PaiNN           | SphereNet       | ForceNet        | GemNet-T        | GemNet-dT       | NequIP          |
> > > > > |:----------------------|:----------------|:----------------|:----------------|:----------------|:----------------|:----------------|:----------------|:----------------|:----------------|
> > > > > | Force                 | $5.8$           | $9.5$           | $1.4$           | $5.1$           | $16.1$          | $10.9$          | $0.7$           | $1.3$           | $1.5$           |
> > > > > | Stability             | $247_{(147)}$   | $232_{(59)}$    | $30_{(10)}$     | $12_{(13)}$     | $500_{(0)}$     | $7_{(3)}$       | $25_{(7)}$      | $7_{(3)}$       | $500_{(0)}$     |
> > > > > | ${RDF}_{(O,O)}$ | $0.07_{(0.01)}$ | $0.63_{(0.04)}$ | $0.27_{(0.15)}$ | $0.30_{(0.14)}$ | $0.89_{(0.04)}$ | $0.79_{(0.03)}$ | $0.22_{(0.05)}$ | $0.42_{(0.22)}$ | $0.06_{(0.02)}$ |
> > > > > | ${RDF}_{(H,H)}$ | $0.06_{(0.02)}$ | $0.30_{(0.02)}$ | $0.18_{(0.08)}$ | $0.21_{(0.09)}$ | $0.40_{(0.01)}$ | $0.55_{(0.01)}$ | $0.16_{(0.03)}$ | $0.35_{(0.25)}$ | $0.05_{(0.01)}$ |
> > > > > | ${RDF}_{(H,O)}$ | $0.19_{(0.05)}$ | $0.57_{(0.04)}$ | $0.21_{(0.04)}$ | $0.29_{(0.12)}$ | $1.14_{(0.03)}$ | $1.34_{(0.03)}$ | $0.20_{(0.04)}$ | $0.42_{(0.27)}$ | $0.27_{(0.07)}$ |
> > > > > | Diffusivity           | $0.04$          | $1.90$          | -           | -           | $2.23$          | -           | -           | -           | $0.18$          |
> > > > > | FPS                   | $91.0$          | $78.9$          | $17.9$          | $71.8$          | $3.1$           | $67.6$          | $11.3$          | $33.7$          | $3.9$           |
> > > > >
> > > > > Water-large (*SphereNet on Water-large requires more memory than Tesla V100. We run its simulation on faster NVIDIA A100 cards, so the FPS is not entirely comparable to other models.):
> > > > > |                       | DeepPot-SE      | SchNet          | DimeNet         | PaiNN           | SphereNet*       | ForceNet        | GemNet-T        | GemNet-dT       | NequIP          |
> > > > > |:----------------------|:----------------|:----------------|:----------------|:----------------|:----------------|:----------------|:----------------|:----------------|:----------------|
> > > > > | Force                 | $10.6$          | $12.1$          | $5.1$           | $9.7$           | $18.4$          | $13.2$          | $5.6$           | $4.2$           | $7.7$           |
> > > > > | Stability             | $19_{(22)}$     | $118_{(58)}$    | $38_{(13)}$     | $16_{(12)}$     | $150_{(0)}$     | $8_{(0)}$       | $45_{(25)}$     | $50_{(9)}$      | $150_{(0)}$     |
> > > > > | ${RDF}_{(O,O)}$ | $0.23_{(0.06)}$ | $0.62_{(0.01)}$ | $0.17_{(0.03)}$ | $0.31_{(0.06)}$ | $0.93_{(0.02)}$ | $0.74_{(0.02)}$ | $0.22_{(0.16)}$ | $0.16_{(0.02)}$ | $0.10_{(0.01)}$ |
> > > > > | ${RDF}_{(H,H)}$ | $0.24_{(0.06)}$ | $0.30_{(0.04)}$ | $0.12_{(0.03)}$ | $0.21_{(0.05)}$ | $0.42_{(0.01)}$ | $0.51_{(0.02)}$ | $0.15_{(0.11)}$ | $0.11_{(0.01)}$ | $0.07_{(0.00)}$ |
> > > > > | ${RDF}_{(H,O)}$ | $0.67_{(0.27)}$ | $0.55_{(0.01)}$ | $0.17_{(0.02)}$ | $0.29_{(0.05)}$ | $0.97_{(0.03)}$ | $1.38_{(0.05)}$ | $0.23_{(0.12)}$ | $0.16_{(0.02)}$ | $0.12_{(0.02)}$ |
> > > > > | Diffusivity           | -           | $2.54$          | -           | -           | $2.98$          | -           | -           | -           | $0.89$          |
> > > > > | FPS                   | $80.7$          | $23.1$          | $3.5$           | $17.4$          | $0.8$           | $11.9$          | $2.2$           | $5.3$           | $0.7$           |
> > > > >
> > > > > As SphereNet uses multi-hop information (angle/torsion) in its message passing, it suffers from scalability and has lower FPS in a condensed-phase system like our water system. This also influences training – given our 1-week training time budget on a Tesla V100 GPU, SphereNet has fewer training epochs compared to more efficient models. Interestingly, SphereNet does not perform well on Force error for the water datasets but is very stable. However, it does not produce accurate ensemble properties. Its RDF curves (in Figure 13) show that the dynamics it learns are stable but incorrect. We hypothesize that the angular and torsional information used in SphereNet are great inductive biases for organic molecules, where such interactions are important. It may not be suitable for condensed phase systems with very simple molecules like water.
> > > > >
> > > > > We are currently running PaiNN and SphereNet experiments on other water dataset sizes, alanine dipeptide, and LiPS. We will update the results once they become available.

---

> ### Comment · Reviewer_NbtG · 2022-11-17
> **Reply to authors**
>
> I appreciate the authors' hard work during the rebuttal. Here are several following questions:
> 1. About OC20/22, and datasets in [1]:
>
> Thanks for pointing out that OC20/22 are NOT molecular dynamics datasets but structural relaxation datasets, and the systems in [1] are at real-world length/time scales in the units of meter and second, but MD simulations are at microscopic length/time scales in the units of nanometer and nanosecond.
>
> I have a question (maybe naive): what is the key difference between MD datasets and other datasets? **Is length/time scales the key difference?**
>
> I have this question because I am familiar with some of these datasets, like MD17 (**MD dataset**), water, goop, etc in [1], and OC20/22 (**not MD dataset**). Based on my understanding, previously, **forces are the common evaluation metric on all of these datasets** (acceleration in [1] is also related to force since F=ma).
>
> In addition, The authors said, **Extending the OC20/22 dataset to study MD requires 2-4 orders of magnitude more compute than OC20/22**. I am wondering how to can we extend the OC20/22 dataset to study MD. Thanks.
>
>
> 2. About GemNet(-T):
>
> Based on my understanding, GemNet(-T/-Q) is SE(3)-invariant method since the inputs to the model are all SE(3)-invariant features (no direction information on edges). If we rotate the given molecule, the input features are always the same. Therefore, I think it is SE(3)-invariant.
>
> And I agree that GemNet(-dT/-dQ) is SE(3)-equivariant since these exist direction information on edges.
>
> [1] Learning to Simulate Complex Physics with Graph Networks

---

> > ### Author Response · Authors · 2022-11-17
> > **Thank you for the reply; Response to follow-up questions**
> >
> > Thank you very much for the reply and for the great questions.
> >
> > > What is the key difference between MD datasets and other datasets? **Is length/time scales the key difference?**
> >
> > We think this question can be answered from two different perspectives: learning and physics.
> >
> > - From the learning perspective, we think the key difference is in the **high complexity of MD interactions and simulation length requirement**. These two factors make learning MD simulation hard. The dynamics governing an MD system are highly non-smooth due to diverse atom types and complex many-body interactions. The non-smooth dynamics also require short time steps, so useful simulation needs to be very long (millions of steps). Datasets in [1] are governed by smoother dynamics and thus allow much longer time steps. The simulation horizon is usually not long (<=1000 steps), and evaluating multi-step errors is sufficient without involving observables. In addition, training ML FF does not require sequential data ordered in time (as energy and forces are instantaneous), while training ML simulators like in [1] requires sequential data.
> >
> > - From the physics perspective, we think the key difference is **an MD system is governed by a potential energy surface** $E$ – a scalar function of the atomistic state. All dynamics and properties can be derived from the potential energy surface, where $\mathbf{F} = -\partial E / \partial \mathbf{x}$. The simulation is done by pairing the potential energy function with the appropriate temperature and thermostats. On the other hand, datasets in [1] do not have a notion of energy, and the dynamics are directly designed to mimic natural phenomena. As an example, an ML FF learned from data generated with one temperature can be used to simulate the system at a different temperature (while the model does not need to be conditioned on the temperature). MD simulations are also time-reversible, and the microscopic motion in MD is typically chaotic, which is extremely sensitive to small perturbations. While macroscopic simulations like those in [1] are usually time irreversible and ignore the fast fluctuations/motions that appeared in microscopic MD simulations.
> >
> > > How can we extend the OC20/22 dataset to study MD?
> >
> > The goal of structural relaxation in OC20/22 is to sample stable conformations. Sampling stable conformations can also be done through MD simulations but would require many more calculations. The energy-minimization-based structural relaxation in OC20/22 is fast in computation but does not contain dynamical information. The relaxation process does not have a notion of time or temperature. Using MD simulation instead will allow us to study the dynamic and give us more diverse samples of stable states, which has been shown in past works [2].
> >
> > > About GemNet(-T).
> >
> > After a closer read, we agree with the reviewer that GemNet(-T/-Q) is SE(3)-invariant (and also invariant to reflection – so E(3)-invariant) as no direction information is used. We have updated the manuscript. Thank you again for correcting us.
> >
> > We are more than happy to further discuss if you have more comments/questions.
> >
> > Reference:
> >
> > [1] Sanchez-Gonzalez, Alvaro, et al. "Learning to simulate complex physics with graph networks." International Conference on Machine Learning. PMLR, 2020.
> >
> > [2] Daelman, Nathan, Marçal Capdevila-Cortada, and Núria López. "Dynamic charge and oxidation state of Pt/CeO2 single-atom catalysts." Nature materials 18.11 (2019): 1215-1221.

---

> > > ### Comment · Reviewer_NbtG · 2022-11-17
> > > **Thanks for the authors' reply**
> > >
> > > Thanks for the authors' reply. My concerns are addressed, and I increased my score.

---

> > > > ### Author Response · Authors · 2022-11-18
> > > > **Thank you for the reply**
> > > >
> > > > Thank you for your reply and for acknowledging that we have addressed your concerns. We have updated more experimental results. May we know if there are still any specific issues that prevent you from recommending acceptance? We are more than happy to address them and further improve the paper.
> > > >
> > > > Thanks!

---

> > > > > ### Comment · Reviewer_NbtG · 2022-12-07
> > > > > **Reply to authors**
> > > > >
> > > > > Firstly, I appreciate the authors' hard work during the rebuttal, adding new experiments and responding to my questions. Most of my concerns are addressed.
> > > > >
> > > > > However, I increased my score from 3 to 5 rather than 6 or higher mainly because of the initial version of this paper. As a paper for **Benchmark and Critical Evaluation**, the datasets, evaluation metrics, and baseline methods in the initial version were unsatisfactory.
> > > > >
> > > > > For example, as shown in my initial comment (https://openreview.net/forum?id=_V-nKeWvs7p&noteId=zSHgBCc2wU0), the task is to predict forces (should be equivariant), but only one equivariant baseline method is included in the initial version of this paper. There are also some unclear or unsatisfactory parts about the datasets and evaluation metrics in the initial version.

---

> > > > > > ### Author Response · Authors · 2022-12-08
> > > > > > **Thank you for the reply**
> > > > > >
> > > > > > Thank you for the reply.
> > > > > >
> > > > > > We want to clarify further that all models except for ForceNet make equivariant force predictions, as the differentiation $-\partial E/ \partial \mathbf{x}$ is equivariant to $\mathbf{x}$ when $E$ is invariant to $\mathbf{x}$. In addition to equivariant predictions, three models in the benchmark (GemNet-dT, NequIP, and PaiNN) use directional features, which are more expressive.
> > > > > >
> > > > > > We believe the purpose of the discussion period is to improve the paper and hope that our submission is evaluated based on its current state, inclusive of all the new experiments and clarifications added during the discussion period, not solely based on the original submission.
> > > > > >
> > > > > > Thanks!

---

> > > > > > > ### Comment · Reviewer_NbtG · 2022-12-09
> > > > > > > **Reply to authors**
> > > > > > >
> > > > > > > Thank you for the reply.
> > > > > > >
> > > > > > > I want to clarify further that my evaluation is based on my understanding of this paper, from the initial version to the current version, not solely based on the original submission.
> > > > > > >  - For the original version, my score was 3.
> > > > > > >  - I increased my score to 5 based on the improvement during rebuttal.
> > > > > > >  - I kept my score as 5, one reason is to let the ACs be aware of my initial comments. Another reason is the Technical Novelty And Significance. I think the technical contributions are only marginally significant or novel. And it seems that most reviewers give a rate of 2 for Technical Novelty And Significance.
> > > > > > >
> > > > > > > Thanks.

---

> ### Author Response · Authors · 2022-11-25
> **More experimental results (1)**
>
> We have finished running experiments of PaiNN and SphereNet on LiPS, Alanine dipeptide, water-1k, and water-90k. We summarize the results below. We can not update the paper as of now and will update the results (Figure 4, Table 5, Table 6, Table 8, Table 9, Figure 9, Figure 12, Figure 14, Figure 15)in the final version.
>
> Overall, both PaiNN and SphereNet are not stable enough for the alanine dipeptide task. On the LiPS dataset, both PaiNN and SphereNet are not the best performing model in force error but can simulate stably with reasonable diffusivity estimation.
>
>
> Alanine dipeptide:
>
> |                     | DeepPot-SE                 | SchNet                       | DimeNet                      | PaiNN                        | SphereNet                    | ForceNet                     | GemNet-T                     | GemNet-dT                    | NequIP                         |
> |:--------------------|:---------------------------|:-----------------------------|:-----------------------------|:-----------------------------|:-----------------------------|:-----------------------------|:-----------------------------|:-----------------------------|:-------------------------------|
> | Force MAE           | $272.1$      | $217.0$       | $239.0$     | $266.2$        | $256.3$  | $284.7$        | $233.5$     | $219.7$       | $215.6$         |
> | #Finished           | 0/6          | 0/6            | 0/6            | 0/6            | 0/6            | 0/6            | 0/6            | 0/6            | 5/6        |
> | Stability           | $0_{(0)}$    | $0_{(0)}$      | $0_{(0)}$      | $0_{(0)}$      | $4_{(7)}$      | $0_{(0)}$      | $18_{(27)}$    | $0_{(0)}$      | $4168_{(1860)}$  |
> | ${PMF}_{\phi}$ | -  | -  | -  | -  | -  | -  | -  | -  | $108_{(2)}$       |
> | ${PMF}_{\psi}$ | -  | -  | -  | -  | -  | -  | -  | -  | $126_{(4)}$       |
> | FPS                 | $54.3$      | $42.4$      | $12.1$         | $42.2$      | $9.9$          | $99.1$        | $15.0$         | $36.5$      | $8.3$            |
>
>
> LiPS:
>
> |             | DeepPot-SE                     | SchNet                          | DimeNet                         | PaiNN                           | SphereNet                       | ForceNet                       | GemNet-T                        | GemNet-dT                       | NequIP                          |
> |:------------|:-------------------------------|:--------------------------------|:--------------------------------|:--------------------------------|:--------------------------------|:-------------------------------|:--------------------------------|:--------------------------------|:--------------------------------|
> | Force       | $40.5$           | $28.8$            | $3.2$          | $11.7$        | $8.3$         | $12.8$       | $1.3$            | $1.4$            | $3.7$          |
> | Stability   | $4_{(3)}$        | $50_{(0)}$       | $48_{(4)}$       | $50_{(0)}$       | $50_{(0)}$       | $26_{(8)}$    | $50_{(0)}$       | $50_{(0)}$       | $50_{(0)}$       |
> | RDF         | $0.27_{(0.15)}$  | $0.04_{(0.00)}$  | $0.05_{(0.01)}$  | $0.04_{(0.01)}$  | $0.04_{(0.00)}$  | $0.51_{(0.08)}$  | $0.04_{(0.00)}$  | $0.04_{(0.00)}$  | $0.04_{(0.01)}$  |
> | Diffusivity | -            | $0.38$        | $0.30$         | $0.40$        | $0.40$        | -            | $0.24$           | $0.28$           | $0.34$         |
> | FPS         | $66.1$          | $35.2$         | $14.8$        | $75.7$           | $18.1$        | $72.1$          | $16.9$        | $43.5$           | $8.2$             |

---

> > ### Author Response · Authors · 2022-11-25
> > **More experiment results (2)**
> >
> > For the water datasets, we see PaiNN is not stable. SphereNet is stable but produces incorrect ensemble statistics. Similar to what we have observed in Water-10k, SphereNet's multi-hop message passing is inefficient for condensed-phased systems, and FPS is low.
> >
> > Water-1k:
> >
> > |                       | DeepPot-SE      | SchNet          | DimeNet         | PaiNN           | SphereNet       | ForceNet        | GemNet-T        | GemNet-dT     | NequIP          |
> > |:----------------------|:----------------|:----------------|:----------------|:----------------|:----------------|:----------------|:----------------|:--------------|:----------------|
> > | Force                 | $6.7$           | $13.1$          | $3.5$           | $5.2$           | $27.5$          | $13.6$          | $5.0$           | $29.2$        | $1.4$           |
> > | Stability             | $108_{(117)}$   | $175_{(56)}$    | $4_{(4)}$       | $14_{(9)}$      | $385_{(160)}$   | $13_{(7)}$      | $6_{(7)}$       | $0_{(0)}$     | $500_{(0)}$     |
> > | ${RDF}_{(O,O)}$ | $0.17_{(0.10)}$ | $0.52_{(0.05)}$ | $0.46_{(0.22)}$ | $0.21_{(0.05)}$ | $0.65_{(0.02)}$ | $0.86_{(0.09)}$ | $0.62_{(0.48)}$ | - | $0.07_{(0.02)}$ |
> > | ${RDF}_{(H,H)}$ | $0.13_{(0.09)}$ | $0.24_{(0.02)}$ | $0.33_{(0.15)}$ | $0.15_{(0.04)}$ | $0.29_{(0.01)}$ | $0.56_{(0.04)}$ | $0.35_{(0.21)}$ | - | $0.07_{(0.02)}$ |
> > | ${RDF}_{(H,O)}$ | $0.28_{(0.15)}$ | $0.54_{(0.01)}$ | $0.43_{(0.17)}$ | $0.16_{(0.04)}$ | $0.81_{(0.03)}$ | $1.44_{(0.09)}$ | $0.71_{(0.65)}$ | - | $0.26_{(0.07)}$ |
> > | Diffusivity           | $0.24$          | $1.79$          | -           | -           | $2.05$          | -           | -           | -         | $0.37$          |
> > | FPS                   | $61.8$          | $99.2$          | $16.4$          | $54.5$          | $3.0$           | $68.1$          | $15.4$          | $34.5$        | $3.9$           |
> >
> > Water-10k:
> >
> > |                       | DeepPot-SE      | SchNet          | DimeNet         | PaiNN           | SphereNet       | ForceNet        | GemNet-T        | GemNet-dT       | NequIP          |
> > |:----------------------|:----------------|:----------------|:----------------|:----------------|:----------------|:----------------|:----------------|:----------------|:----------------|
> > | Force                 | $5.9$           | $8.4$           | $1.7$           | $5.1$           | $14.8$          | $8.6$           | $0.7$           | $1.1$           | $1.4$           |
> > | Stability             | $500_{(0)}$     | $299_{(70)}$    | $36_{(9)}$      | $16_{(7)}$      | $500_{(0)}$     | $9_{(12)}$      | $20_{(9)}$      | $8_{(10)}$      | $500_{(0)}$     |
> > | ${RDF}_{(O,O)}$ | $0.07_{(0.02)}$ | $0.67_{(0.03)}$ | $0.21_{(0.03)}$ | $0.26_{(0.17)}$ | $0.91_{(0.04)}$ | $1.31_{(0.49)}$ | $0.35_{(0.23)}$ | $0.20_{(0.01)}$ | $0.06_{(0.01)}$ |
> > | ${RDF}_{(H,H)}$ | $0.05_{(0.01)}$ | $0.31_{(0.02)}$ | $0.14_{(0.01)}$ | $0.20_{(0.12)}$ | $0.42_{(0.03)}$ | $0.82_{(0.26)}$ | $0.25_{(0.19)}$ | $0.16_{(0.01)}$ | $0.04_{(0.01)}$ |
> > | ${RDF}_{(H,O)}$ | $0.29_{(0.08)}$ | $0.67_{(0.04)}$ | $0.18_{(0.02)}$ | $0.21_{(0.06)}$ | $1.24_{(0.08)}$ | $2.05_{(0.60)}$ | $0.24_{(0.06)}$ | $0.26_{(0.02)}$ | $0.25_{(0.06)}$ |
> > | Diffusivity           | $0.35$          | $1.97$          | -           | -           | $2.26$          | -           | -           | -           | $0.18$          |
> > | FPS                   | $62.1$          | $103.0$         | $16.3$          | $71.9$          | $3.1$           | $43.8$          | $15.3$          | $32.7$          | $3.0$           |

---

### Official Review · Reviewer_UiVb · 2022-10-31

**Confidence:** 4
**Correctness:** 3
**Technical Novelty And Significance:** 3
**Empirical Novelty And Significance:** 3
**Recommendation:** 6

**Clarity, Quality, Novelty And Reproducibility:**

Clearly, it is a benchmarking and analysis paper in AIMD. The major novelty of this paper is the provision of a suite with baseline models, datasets, and evaluation metrics.

**Strength And Weaknesses:**

Strengths:
1. The comparison and analysis in this paper is through. Insights into failure cases are provided.
2. The efforts of providing a benchmarking suite should be praised, considering that ML MD is a new area in AI for chemoinformatics and this is a rather challenging task.

Weaknesses:
1. The size of each simulation systems is restricted to have less than 200 atoms. This restriction may limit future scalability studies.
2. The origins of these four datasets are unclear. The authors should explicitly mention whether each data set was generated from experiments or in silico simulation.
3. Mathematical MD simulation tools such as GROMACS and Amber may be included in comparison if a data set was generated from experiments.

**Summary Of The Paper:**

The authors of this paper propose a benchmark suite for ML MD. Within the suite, 7 deep learning based methods are included as base models and 4 datasets of different MD systems were tailored for simulation. From intensive comparison and analysis, the authors claim that force error is not a sufficient measure to evaluate ML MD performance, and stability should be considered in benchmarking.

**Summary Of The Review:**

Overall, I think this work is very informative and useful for researchers in the area of AI for chemoinformatics.

---

> ### Author Response · Authors · 2022-11-11
> **Response part 1**
>
> We thank reviewer UiVb for helpful feedback and comments. We address each of the reviewer’s concerns below.
>
> > The size of each simulation systems is restricted to have less than 200 atoms. This restriction may limit future scalability studies.
>
> We agree with the reviewer that scalability is an important aspect to consider. In the updated draft, we added a new benchmark dataset with 512 water molecules (1536 atoms) simulated with the same reference force field (SPC/E-fw).
> We evaluate ML FF models trained on water-10k to simulate the 512 water molecules for 150 ps, and compute the same metrics as the small water (64 molecules) dataset: force error, stability, RDF, and diffusivity. **We add a paragraph of discussion and Table 11 to Appendix C where we demonstrate the experiment results.** We also summarize this result in the table below. One standard deviation for applicable metrics is in subscript. The computation of diffusivity requires 100 ps stable simulation.
>
> We observe that all models suffer slightly higher force errors compared to evaluation over the 64-molecule water system. In terms of stability, NequIP always remains stable for the entire 150 ps. SchNet is the second most stable model, while all other models are not stable enough for diffusivity computation. DimeNet, GemNet, and GemNet-dT are not stable throughout the entire simulation but can produce decent RDF results.
>
> Nov 18: results are now updated with SphereNet included. (*SphereNet on Water-large requires more memory than Tesla V100. We run its simulation on faster NVIDIA A100 cards, so the FPS is not entirely comparable to other models.):
> |                       | DeepPot-SE      | SchNet          | DimeNet         | PaiNN           | SphereNet*       | ForceNet        | GemNet-T        | GemNet-dT       | NequIP          |
> |:----------------------|:----------------|:----------------|:----------------|:----------------|:----------------|:----------------|:----------------|:----------------|:----------------|
> | Force                 | $10.6$          | $12.1$          | $5.1$           | $9.7$           | $18.4$          | $13.2$          | $5.6$           | $4.2$           | $7.7$           |
> | Stability             | $19_{(22)}$     | $118_{(58)}$    | $38_{(13)}$     | $16_{(12)}$     | $150_{(0)}$     | $8_{(0)}$       | $45_{(25)}$     | $50_{(9)}$      | $150_{(0)}$     |
> | ${RDF}_{(O,O)}$ | $0.23_{(0.06)}$ | $0.62_{(0.01)}$ | $0.17_{(0.03)}$ | $0.31_{(0.06)}$ | $0.93_{(0.02)}$ | $0.74_{(0.02)}$ | $0.22_{(0.16)}$ | $0.16_{(0.02)}$ | $0.10_{(0.01)}$ |
> | ${RDF}_{(H,H)}$ | $0.24_{(0.06)}$ | $0.30_{(0.04)}$ | $0.12_{(0.03)}$ | $0.21_{(0.05)}$ | $0.42_{(0.01)}$ | $0.51_{(0.02)}$ | $0.15_{(0.11)}$ | $0.11_{(0.01)}$ | $0.07_{(0.00)}$ |
> | ${RDF}_{(H,O)}$ | $0.67_{(0.27)}$ | $0.55_{(0.01)}$ | $0.17_{(0.02)}$ | $0.29_{(0.05)}$ | $0.97_{(0.03)}$ | $1.38_{(0.05)}$ | $0.23_{(0.12)}$ | $0.16_{(0.02)}$ | $0.12_{(0.02)}$ |
> | Diffusivity           | -           | $2.54$          | -           | -           | $2.98$          | -           | -           | -           | $0.89$          |
> | FPS                   | $80.7$          | $23.1$          | $3.5$           | $17.4$          | $0.8$           | $11.9$          | $2.2$           | $5.3$           | $0.7$           |
>
> We note that the large water system is very expensive to simulate. It requires 62 GPU hours to simulate a single 150 ps trajectory using NequIP on a Tesla V100 GPU. The main reason that we choose small systems for benchmarking is the high computational cost for MD simulation, as we wish to establish computationally accessible benchmarks to diverse users. Simulations reported in this paper take up to 75 hours for a single trajectory, as millions of steps are required to obtain meaningful results. The total GPU hours used in the paper’s experiments (not including training the models) exceeds 3,000 GPU hours. In addition, training the models uses 1000+ GPU hours. Despite small sizes, the systems included in this paper are carefully chosen to be representative of a variety of real-world MD applications.

---

> > ### Author Response · Authors · 2022-11-11
> > **Response part 2**
> >
> > > The origins of these four datasets are unclear. The authors should explicitly mention whether each data set was generated from experiments or in silico simulation.
> >
> > All datasets included in this paper are obtained through in silico simulations. With existing technology, MD trajectories cannot be obtained from experiments given the extremely small spatial/time scales. Experiment measurements are instead often used to guide the development of in silico simulation tools, such as AMBER.
> > We include more details on the origins of the datasets in Appendix B and briefly explain them here:
> >
> > MD17: The MD17 dataset is generated from path-integral molecular dynamics simulations that incorporate quantum mechanics into the classic molecular dynamics simulations using Feynman path integrals. These simulations are at the ab-initio level. More details can be found in the original paper [1].
> >
> > Water: Both small and large, The water simulations use classical molecular dynamics methods, where atoms are explicitly described with electrons interactions approximated by potential energies. We use a modified version of the simple-point charge water model where the interaction parameters (e.g., O-H bond stretch and H-O-H bond angles) are parameterized to match extensive experimental properties such as the self-diffusion and dielectric constants at the bulk phase.
> >
> > Alanine Dipeptide: The alanine dipeptide simulations are conducted using the AMBER-03 force field at the classical molecular dynamics level. In the AMBER-03 force field, the potential energy parameters such as van der Waals and electrostatics are mostly derived from quantum mechanical methods with minor optimization on the bonded parameters to reproduce the experimental vibrational frequencies and structures [2,3].
> >
> > LiPS: The LiPS datasets are generated by ab-initio molecular dynamics simulations with a generalized gradient PBE functional and projector augmented wave pseudopotentials. More details can be found in the original paper [4].
> >
> > > Mathematical MD simulation tools such as GROMACS and Amber may be included in comparison if a data set was generated from experiments.
> >
> > For all datasets considered, Mathematical MD simulations based on quantum/classical calculations are used to generate the reference data.
> >
> >
> > We look forward to further discussion if you have additional feedback.
> >
> > Reference:
> >
> > [1] Chmiela, Stefan, et al. "Machine learning of accurate energy-conserving molecular force fields." Science advances 3.5 (2017): e1603015.
> >
> > [2] Cornell, Wendy D., et al. "A second generation force field for the simulation of proteins, nucleic acids, and organic molecules J. Am. Chem. Soc. 1995, 117, 5179− 5197." Journal of the American Chemical Society 118.9 (1996): 2309-2309.
> >
> > [3] Ponder, Jay W., and David A. Case. "Force fields for protein simulations." Advances in protein chemistry 66 (2003): 27-85.
> >
> > [4] Batzner, Simon, et al. "E (3)-equivariant graph neural networks for data-efficient and accurate interatomic potentials." Nature communications 13.1 (2022): 1-11.

---

### Decision · Program_Chairs · 2023-01-20

**Decision:**

Reject

**Justification For Why Not Higher Score:**

Some reviewers are only moderately supportive to this work, and one reviewer is still not convinced with the significance.

**Justification For Why Not Lower Score:**

NA

**Metareview: Summary, Strengths And Weaknesses:**

This work provides a benchmark evaluation for machine learning methods in force field prediction in molecular simulation. After review and quite intensive discussions, most reviewers are only moderately supportive to this work, and one reviewer is still not convinced with the significance. We believe this work has some level of values, but might not be a good fit to ICLR. We are recommending to reject this work and encourage the authors to consider NeurIPS datasets and benchmark track, which would be a better fit.

**Summary Of Ac-Reviewer Meeting:**

A meeting was hold among the AC and a subset of reviewers. Some reviewers are only moderately supportive to this work, and one reviewer is still not convinced with the significance.